# Evolutionary safety of lethal mutagenesis driven by antiviral treatment

**Gabriela Lobinska**[1], **Yitzhak Pilpel**[1]*, **Martin A. Nowak**[2]*

1 Department of Molecular Genetics, Weizmann Institute of Science, Rehovot, Israel, 2 Department of Mathematics, Department of Organismic and Evolutionary Biology, Harvard University, Cambridge, Massachusetts, United States of America

* pilpel@weizmann.ac.il (YP); martin_nowak@harvard.edu (MAN)

**Data Availability Statement:** Our code for mathematical simulations is available at https://github.com/gabriela3001/molnupiravir_evol_safety, DOI: 10.5281/zenodo.8017992. No biological data were generated during this project.

## Abstract

Nucleoside analogs are a major class of antiviral drugs. Some act by increasing the viral mutation rate causing lethal mutagenesis of the virus. Their mutagenic capacity, however, may lead to an evolutionary safety concern. We define evolutionary safety as a probabilistic assurance that the treatment will not generate an increased number of mutants. We develop a mathematical framework to estimate the total mutant load produced with and without mutagenic treatment. We predict rates of appearance of such virus mutants as a function of the timing of treatment and the immune competence of patients, employing realistic assumptions about the vulnerability of the viral genome and its potential to generate viable mutants. We focus on the case study of Molnupiravir, which is an FDA-approved treatment against Coronavirus Disease-2019 (COVID-19). We estimate that Molnupiravir is narrowly evolutionarily safe, subject to the current estimate of parameters. Evolutionary safety can be improved by restricting treatment with this drug to individuals with a low immunological clearance rate and, in future, by designing treatments that lead to a greater increase in mutation rate. We report a simple mathematical rule to determine the fold increase in mutation rate required to obtain evolutionary safety that is also applicable to other pathogen-treatment combinations.

## Introduction

Nucleoside analogs are molecules similar in shape to naturally occurring nucleosides used by living organisms and viruses for nucleic acid synthesis. They are therefore readily incorporated into nascent DNA or RNA chains by viral polymerases. Many nucleoside analogs differ from natural nucleosides in key aspects which usually prevents further viral genome chain elongation. Some nucleoside analogs lack a 3'OH group which makes the viral polymerase unable to attach the next nucleoside to the growing chain. Others, such as Lamivudine, cause steric hindrance upon incorporation into the DNA or RNA chain [1–3].

Other nucleoside analogs do not prevent viral RNA elongation. Instead, they have the capacity to ambiguously base pair with several nucleosides, causing erroneous incorporation of nucleosides during the replication process. Thereby such drugs increase the virus mutation

**Funding:** This work was supported by the Kimmel Foundation (YP) https://www.weizmann.ac.il/acadaff/awards-and-honors/institute-honors-and-prizes/Kimmel; and the Minerva Center on Live Emulation of Evolution in the Lab (YP) https://www.minerva.mpg.de/centers/list/minerva-center-on-live-emulation-of-evolution-in-the-lab. YP holds the Ben May Professorial Chair. The funders had no role in study design, data collection and analysis, decision to publish, or preparation of the manuscript. The authors did not receive any salary from any of the funders.

**Competing interests:** The authors have declared that no competing interests exist.

**Abbreviations:** COVID-19, Coronavirus Disease-2019; ERF, evolutionary risk factor; HIV, human immunodeficiency virus; IRF, infectiousness risk factor; SARS-CoV-2, Severe Acute Respiratory Syndrome Coronavirus 2; VoC, variants of concern; VSV, vesicular stomatitis virus.

rate up to the point of lethal mutagenesis, a mechanism with foundations in quasispecies theory. This theory describes populations of replicating genomes under mutation and selection [4–9]. In addition to increasing the probability of lethal mutations, mutagenic treatment can also decrease the number of viable virions through lethal defection [10]. Lethal defection occurs when functional proteins synthetized from viable viral genomes are consumed for the packaging of defective ones that coexist in same cell.

Repurposing mutagenic antiviral drugs to treat Coronavirus Disease-2019 (COVID-19) has been suggested early on in the pandemic [11]. Molnupiravir, a prime example, seems to act exclusively through mutagenesis. Its incorporation into nascent RNA genomes by the viral polymerase does not result in chain termination, in fact, the viral RNA polymerase has been shown to successfully elongate RNA chains after the incorporation of Molnupiravir [12–14]. Molnupiravir switches between 2 tautomeric forms: one is structurally similar to a cytosine, the other is structurally similar to a uracil. Hence, Molnupiravir can base pair, depending on its form, either with guanosine or with adenosine [12,13]. Severe Acute Respiratory Syndrome Coronavirus 2 (SARS-CoV-2) is a positive-sense single-stranded RNA virus and its RNA replication proceeds in 2 steps. First, the negative-sense RNA is polymerized based on the plus strand, and the negative strand then serves as a template to synthetize positive-sense RNA molecules [15]. Hence, the incorporation of Molnupiravir during the first step of RNA synthesis gives rise to an ambiguous template: positions where Molnupiravir was incorporated can be read by the RNA-dependent RNA polymerase as either guanosine or adenosine. This causes mutations in the progeny RNA compared with the parental RNA, possibly up to the point of the "error threshold" and death of the virus [12–14], see **Fig 1A**. For a discussion of error threshold and lethal mutagenesis, see [6,7,16–20].

As noted before, the intended antiviral activity of Molnupiravir resides in its capacity to induce mutagenesis and hence reduce virus load. Yet, this very property which confers to Molnupiravir its desired antiviral effect might also enhance the capacity of the virus to develop drug resistance, immune evasion, infectivity, infectiousness, or other undesired phenotypes. Thus, a mathematical analysis should weigh the desired and potentially deleterious effects of mutagenesis drugs in general and of the present virus and drug in particular.

Mathematical theory has established the concept of an error threshold, which is the maximum mutation rate compatible with adaptation (or survival) of a population of replicating agents [9,21,22]. But a theoretical analysis is still missing to evaluate the risk of emergence of variants of concern (VoC) or simply viable mutants following mutagenic treatment. In the context of the COVID-19 pandemic, the mutagenic potential of Molnupiravir leads to concerns about accelerating SARS-CoV-2 evolution. VoC can include mutants that are resistant against vaccination or antiviral treatments as well as mutants with enhanced transmissibility or lethality.

Since the beginning of the COVID-19 pandemic, mathematical modeling has been used to establish vaccination strategies minimizing the risk of emergence of resistant mutants [23–27], predict the epidemiological spread of SARS-CoV-2 [28–32], and optimize the guidelines concerning isolation of contact and positive cases [33,34]. The emergence of resistance against treatments has been investigated for other viruses, including human immunodeficiency virus (HIV) [35,36], influenza A [37–40], and hepatitis C [41].

Molnupiravir has been found effective in inhibiting the replication of SARS-CoV-2 in ferrets, mice, and cultured human cells [42,43]. Following these promising results, Phase 2 and then Phase 3 clinical trials were conducted and concluded that Molnupiravir is safe and reduces the risk of hospitalization or death by about 50% [44–47]. The recommended dosage is 800 mg twice daily, for 5 days, and within 5 days of symptom onset [48]. It is especially recommended for individuals at high risk for disease progression to severe symptoms and death.

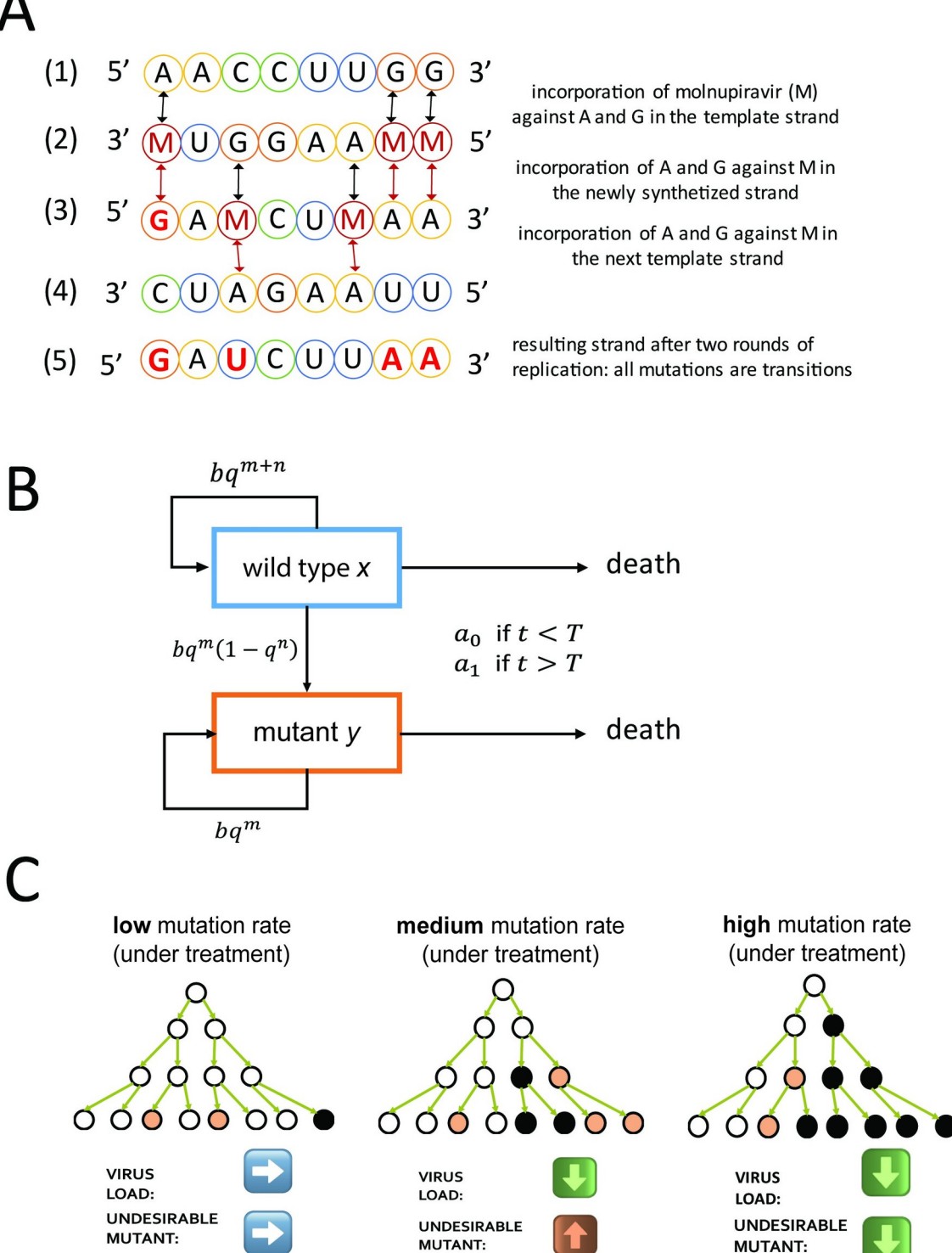

**Fig 1.** (A) Mechanism of action of molnupiravir. SARS-CoV-2 has a positive-sense single-stranded RNA genome, represented schematically in (1). Its replication proceeds in 2 steps: first, the synthesis of a negative-sense template strand (2), which is then used to synthesize a positive-sense progeny genome (3). Molnupiravir (M) is incorporated against of A or G during the synthesis of the negative-sense template strand (2). When the template strand is replicated, M can be base-paired with either G or A. Hence, all A and G in the parent genome become ambiguous and can appear as A or G in the newly synthetized positive-strand genome. C and T are not affected by molnupiravir during the synthesis of the template strand, (1) to (2), but can be substituted to M during the synthesis of

the progeny genome from the template strand, (2) to (3). M can then base-pair with A or G when used as a template; see (3) to (4), which can cause A->U and U->A transitions in the final progeny genome (5). **(B) Virus dynamics within an infected person.** Wild type ($x$) and mutant ($y$) replicate at rate $b$ and quality $q = 1-u$. The per base mutation rate, $u$, is increased by treatment with molnupiravir. Both wild type and mutant need to maintain $m$ positions to remain viable. Mutating any of $n$ positions in the wild type results in a mutant. In the beginning of the infection, the adaptive immune response is weak, and virus is cleared at a rate $a_0$ which is less than $b$. After some time, $T$, the adaptive immunity is strong, and virus is cleared at the rate $a_1$ which is greater than $b$. **(C) Graphical summary of the influence of mutagenic drugs on virus mutants.** White circles represent wild type, beige circles viable mutant, and black circles dead virus. When the mutation rate is low, few viable mutants and few lethal mutants are produced. Most mutations occur when the virus load is already high; hence, they have little influence on subsequent generations. For intermediate mutation rate, the total virus load declines but the amount of viable mutant increases. When the mutation rate is high, both the virus load and the amount of viable mutant decline. SARS-CoV-2, Severe Acute Respiratory Syndrome Coronavirus 2.

While drug's physiological safety is a cornerstone of pharmacology, we explore here a new aspect of drug safety. We define a treatment as "evolutionarily safe" if its use does not increase the rate of generation of mutants in a treated patient beyond the risk expected in an untreated patient. This notion is especially relevant for drugs that work through lethal mutagenesis.

In this paper, we analyze the case study of the increase of the evolutionary potential of a virus (here: SARS-CoV-2) under mutagenic treatment (here: Molnupiravir treatment). In particular, we ask if the wanted effect of limitation of virus load by the drug could be accompanied by an unwanted enhancement in the rate of appearance of viable mutants or VoC due to increased mutagenesis. We construct a mathematical framework describing the increase and decrease of the virus load after infection and derive expressions for the total amount of wild type and mutant produced by individuals during the course of an infection. We use empirical data on COVID-19 and bioinformatic data on SARS-CoV-2 to estimate key parameters, including infection progression within the body amidst response of the immune system and the number of potentially lethal positions in the genome.

We find that the Molnupiravir-SARS-CoV-2 treatment is situated in a region of the parameter space that is estimated to be narrowly evolutionarily safe. Evolutionary safety is expected to increase in patients with decreased immunological viral clearance rate. Our analysis also shows that evolutionary safety increases with the number of positions in viral genome positions that are lethal when mutated. Crucially, evolutionary safety could be improved by obtaining higher increases in the mutation rate under treatment that provides a clear direction for future drug improvement. We derive a simple mathematical formula that determines the evolutionary safety of a drug given the pathogen's mutation rate with and without treatment and the number of positions in the pathogen's genome that are lethal when mutated.

## Description of the model

After infection with SARS-CoV-2, virus load increases exponentially until it reaches a peak after a median of about 5 days [49]. During this growth phase, the action of the immune system is insufficient to counterbalance viral replication. Subsequently, the immune response gains momentum and infection enters a clearance phase. Now virus load decreases exponentially until the virus becomes eliminated about 10 to 30 days after initial infection [49,50]. In some immunocompromised individuals, viral clearance can take many weeks [51,52]. However, some argue that the isolation of infectious virus is rare after 20 days postinfection [53]. The values for the time to peak of the virus load, time to clearance, and the virus load at peak from several sources are summarized in **Table A1** in **S1 Text**. We provide an overview of related published literature, in the **S1 Text** (see section "Relationship to previous literature").

In our mathematical formalism, we describe the evolution of a virus within the body of a single human host by following the abundance of 2 viral types: wild type, $x$, and mutants, $y$. Both $x$ and $y$ replicate with birth rate $b$ and replication quality $q = 1-u$, where $u$ is the mutation

rate per base. The mutation rate can be altered by the administration of a mutagenic drug. The virus genome contains $m$ positions, all of which must be maintained without mutations in order to generate viable progeny. In addition to those, we consider $n$ positions, such that even a single mutation in one of them gives rise to a mutant virion, $y$.

In this paper, a mutant virion can be any mutant whose emergence we wish to prevent. If we are concerned about a specific VoC, then $n = 1$. If we are concerned about the set of all possible VoCs, then $n>1$, but $n<L-m$, where $L$ is the length of the genome. Lastly, if we are concerned about any viable mutant, then $n = L-m$. In the course of our analysis, we found that the value of $n$ has little effect on the evolutionary safety of a mutagenic treatment.

As common in mutagenesis and also in the specific mechanism of action of Molnupiravir, transition mutations are more likely than transversion mutations (see **Fig 1A**). Our model can be extended to consider situations where the mutagenic drug increases the probability of mutation for a subset of all possible mutations (see **Methods**). Both $x$ and $y$ are cleared at same rate $a_j$ with the subscript $j$ indicating the presence or absence of an adaptive immune response, such that during the growth phase $j = 0$ and during the clearance phase $j = 1$. We have $a_0<b<a_1$. Virus dynamics [16] in an infected patient can be described by the system of differential equations

$$\dot{x} = x(bq^{m+n} - a_j)$$

$$\dot{y} = xbq^m(1 - q^n) + y(bq^m - a_j). \qquad (1)$$

We ignore back mutation from mutant to wild type [4,7,16]. In the growth phase, without treatment, we have $bq^{m+n}>a_0$ since both $x$ and $y$ grow exponentially. In the clearance phase, without treatment, we have $bq^m<a_1$ since both $x$ and $y$ decline exponentially. The system is linear and can be solved analytically (see **Methods**). The biological reactions are presented schematically in **Fig 1B**. In our simple approach, there is a sharp onset of adaptive immunity that happens at time $T$. We relax this assumption in a model extension (see section "Gradual activation of the immune system").

## Values of parameters

All parameters and sources for their values are summarized in **Table 1**. Each parameter can be found in, or calculated based on, the existing literature.

## Mutation rates

We denote by $u_0$ the mutation rate without mutagenic treatment and by $u_1$, which is greater than $u_0$, the mutation rate with mutagenic treatment.

The typical mutation rate for other positive single-strand RNA viruses is $10^{-5}$ per base [54]. The mutation rate of SARS-CoV-2 has been hypothesized to be lower because of a proofreading capability [55]. The per-base mutation rate has been estimated at $u_0 = 10^{-6}$ per bp by proxy with the related beta-coronavirus MHV [56,57]. An in vitro study of experimental evolution of SARS-CoV-2 has reached the estimate $u_0 = 3.7 \cdot 10^{-6}$ per base [58]. Zhou and colleagues [14] estimated the mutation rate in vitro of SARS-CoV-2 to be closer to $10^{-5}$ per base. Although the mode of replication (i.e., stamping versus linear replication [59]) and the number of mutations in the plus/minus strands can affect the distribution of the number of mutants, we found that it does not affect their expected number. For a detailed analysis of the influence of the mode of replication, the mutation rate in the plus/minus strand and RNA editing on the mutation rate,

**Table 1. Summary of parameters with ranges for their values and method of estimation.**

| Symbol | Name | Value | Method of approximation | References |
|---|---|---|---|---|
| $b$ | Birth rate of infected cells | 7.61 | Fitted to virus load along time measurements in infected patients | [50,73–76] |
| $a_0$ | Clearance rate prior to adaptive immune response | 3 | Computed from eclipse time of SARS-CoV-2 in infected cells in vitro | [57] |
| $a_1$ | Clearance rate during adaptive immune response | 7.7–10 | Fitted to virus load along time measurements in infected patients | [50,73–76] |
| $u_0$ | Viral mutation rate without treatment | $10^{-6}$ - $5 \cdot 10^{-5}$ | Mutation rate measured for related MHV | [14,56–58] |
| $u_1$ | Viral mutation rate during treatment | $2-5 \cdot 10^{-6}$ | Fold increase in mutation rate under treatment measured in treated patients and in vitro | [14,20,44] |
| $m$ | Number of lethal positions in SARS-CoV-2 genome | ~12,000 | Typical proportion of lethal mutations in ssRNA viruses | [65,66] |
| | | ~21,000 | Typical proportion of lethal + severely deleterious mutations in ssRNA viruses | |
| $n$ | Number of beneficial positions in SARS-CoV-2 genome | ~100 | Analysis of mutagenesis data | [67,68] |
| $T$ | Time of peak of virus load | 3–7 | Virus load along time measurements in infected patients | [50,73–76] |

SARS-CoV-2, Severe Acute Respiratory Syndrome Coronavirus 2.

see section in File S3 "Estimating the mutation rate." For our main analysis, we use $u_0 = 10^{-6}$ per base. We also explore results for $u_0 = 5 \cdot 10^{-6}$ and $u_0 = 10^{-5}$.

The mutation rate of SARS-CoV-2 under Molnupiravir treatment has been measured in vitro to be 2- to 5-fold higher than without treatment [14]. The fold increase in mutation rate under treatment can also be estimated from sequencing viral samples from treated patients. A 2-fold increase in the mutation rate in RNA-dependent RNA polymerase sequence in patients treated with Molnupiravir has been observed during its Phase 2a clinical trial [44]. This estimate comes with the caveat of neglecting potentially rare, severely deleterious mutants since those are less likely to be sequenced. Hence, we estimate $u_1$ to be 2 to 5 times higher than $u_0$, relying on the in vitro estimate. Mutation rate estimations for different pathogen–drug combinations are available in the literature and result in even higher estimates for the virus mutation rate under treatment [20]. In our analysis, we explore a wide range of $u_1$ values, because it is our expectation that future mutagenic treatments might achieve higher increases of the virus mutation rate.

## Viral birth and clearance rates

The average lifetime of SARS-CoV-2 has been measured by proxy with MHV in monkey kidney cells and was found to be about 8 hours [57]. Hence, without infection of new cells, we would obtain a clearance rate of $a_0 = 3$ per day. From the current literature, we know that the virus load grows by about 10 orders of magnitude within 5 days [49,60]. Hence, for the viral growth rate we obtain $b = 7.61$ per virion per day. For the clearance phase, a decrease by 4 orders of magnitude in 10 days results in a death rate of $a_1 = 8.76$ per day reflecting high immunocompetence. The same fold decrease over 120 days results in a death rate of $a_1 = 7.69$ per day reflecting low immunocompetence (see **Methods**). These estimates are approximations as they ignore loss by lethal mutants.

## Number of viral genome positions that are either lethal or potentially concerning when mutated

The fitness effects of some individual mutations has been studied for some viruses such as the influenza virus [61,62], the HIV [63], or hepatitis C virus [64]. However, the number of lethal mutations $m$ and the number of mutations that are potentially concerning needs to be

computed for the whole SARS-CoV-2 genome. The distribution of fitness effects of random, single mutations has been studied in a different single-stranded RNA virus, the vesicular stomatitis virus (VSV) [65]. This distribution seems to be similar among single-stranded RNA viruses but could differ between species [66]. According to these studies, the proportion of mutations that are lethal when mutated is about 40% and the proportion of highly deleterious mutations, defined as those that reduce the viral fitness by more than 25%, represents about 30% of possible mutations. Note that the low mutation rate that is characteristic of SARS-CoV-2 allows us to approximate the number of lethal positions as 1/3 of the total number of possible mutations, taking into account that each position can be mutated to 3 different destinations (see **Methods**). SARS-CoV-2 genome has a length of 29,900 nucleotides. Hence, assuming 40% of positions being lethal upon mutation, we have $m$ = 11,960 positions when considering lethal mutations only and $m$ = 20,930 positions when considering that 70% of positions are either lethal or highly deleterious upon mutations. Hence, the realistic range for $m$ is between 11,960 and 20,930 positions. For completeness, we also explore unrealistically lower bound of $m$ such as 1,500 positions, which is the number of positions in the coding genome that are one nucleotide way from a STOP codon, assuming that most nonsense mutations are deleterious or lethal.

We first consider mutants that are VoCs, which is exhibit phenotypes such as for example increased infectiousness or virulence. In order to estimate the number of positions that could give rise to new variants of concern when mutated (denoted by $n$), we used empirical data collected by [67,68]. Starr and colleagues conducted deep mutagenesis scans of the receptor-binding domain of the SARS-CoV-2 spike protein. For each of the generated mutants, Starr and colleagues measured the mutant's binding affinity to ACE2 that is the receptor used by SARS-CoV-2 to enter the human cell. In a subsequent study, Starr and colleagues also measured each mutant's affinity to antibodies in order to assess the ability of each mutant to escape the adaptive immune response and antibody treatments. Both escape from antibody and increased affinity to ACE2 are phenotypes beneficial for SARS-CoV-2 and are hence concerning. We identified 484 amino acid substitutions that result in antibody escape and 314 distinct amino acid substitutions that result in increased binding to ACE2. For each position coding for the receptor-binding domain of the spike protein, we counted how many mutations can give rise to the identified set of substitutions with either increased binding to ACE2 or decreased binding to antibodies (we corrected for the overlap of substitutions found in both categories). We found that the resulting estimate (divided by 3 to take into account all possible destinations, see **Methods**) was $n$ = 87 positions when considering all possible mutations and $n$ = 75 positions when considering only transition mutations, i.e., when taking into account the specific mechanism of action of Molnupiravir.

Of course, mutations that are advantageous for the virus could occur also outside of the receptor-binding domain of the spike protein. More broadly, any neutral and even slightly deleterious mutation can be undesirable since they could represent an evolutionary "stepping-stone" to a multiple-mutation variant due to epistasis. Hence, we also explore how considering a very large number of positions that could be potentially concerning when mutated, up to the length of the SARS-CoV-2 genome minus the $m$ positions that are lethal when mutated. Thus, our upper bound on the value of $n$ is 29,900−$m$. In addition, we explore the possibility of double mutants in the section "Evolutionary safety for higher-order mutants."

## Abundance of mutant virus for various treatment regimes

In **Fig A1** in **S1 Text**, we show the dynamics of total virus and mutant over the course of an infection. We consider 4 times for the start of mutagenic treatment: at infection; at day 2 after

infection, which corresponds to the beginning of symptoms; at day 5 after infection, which corresponds to the peak of the virus load; and at day 7 after infection. We observe that under each of these 4 options, treatment decreases the abundance of wild-type virus. The dynamics of mutant follows that of the wild type. For the parameters used in **Fig A1** in **S1 Text**, treatment decreases the abundance of mutant virus—with exception of a brief transient period soon after the start of therapy, which is almost invisible in the figure.

In **Fig A2** in **S1 Text**, we assess the plausibility of the model by plotting virus load versus time for different values of the death rate during the clearance phase and comparing it to sequential measurements of virus load in patients. In **Table A2** in **S1 Text**, we use measurements of virus load from patients that were treated or not with Molnupiravir.

We are interested in calculating the total number of mutant virus produced over the course of infection. This number can be computed as the integral of the abundance of mutant virus over time (see **Methods**). We consider 2 scenarios: in the first, the patient begins treatment when their virus load reaches its peak; in the second, the patient begins treatment when they become infected (following exposure to an infected individual). Note that even without mutagenic treatment, due to the innate mutation rate, viral mutants will appear. Thus, our aim is to evaluate their total abundance for various mutation rates, with and without treatment.

## Treatment begins at (or near) peak virus load

In **Fig 2**, we show the cumulative mutant load, $Y(u_1)$, as a function of the mutation rate $u_1$ for the case where treatment starts at peak virus load. To understand this function, we introduce the parameter $\eta = x_T/y_T$, with $x_T$ and $y_T$ denoting, respectively, wild-type and mutant virus load at peak. If $\eta > n/m$ then $Y(u_1)$ is a declining function. In this case, any mutagenic treatment is evolutionarily safe in the sense of reducing the cumulative mutant virus load compared to no treatment. If $\eta < n/m$ then the function $Y(u_1)$ attains a single maximum at

$$u^* = \frac{a_1 - b}{mb} \frac{n - \eta m}{n + \eta m}. \tag{2}$$

If $u_1 > u^*$ then any increase in mutation rate is beneficial for evolutionary safety as it actually *decreases* the chance of appearance of potentially concerning mutants compared to evolution of the virus under no treatment. If $u_1 < u^*$ then a small increase in the mutation rate can increase the chance of appearance of potentially concerning mutants under treatment, and thus be evolutionarily unsafe; in this case, there needs to be a sufficiently large increase in mutation rate to make the treatment evolutionarily safe (see **Fig 2** for details). We notice that increasing estimates of $m$ or decreasing $a_1$ reduces the value of $u^*$ and therefore increases the range of $u_0$ for which mutagenic treatment is evolutionarily safe. In particular, the slower the patient clears the virus (lower $a_1$), the lower the value of $u^*$ and hence treatments become more evolutionarily safe. In **Fig 2**, we notice that only for low $m$ and high $a_1$ we find $u^* > u_0$. For all other cases, $u^* < u_0$, and lethal mutagenesis is both evolutionarily safe and desired, because it reduces the abundance of both wild type and mutant.

If the mutagenic treatment is strong enough, it precludes the replication of the virus. Hence, it is always safe. We indicate, with a red arrow, the error threshold for the virus (see **Fig 2**). If the mutation rate exceeds the error threshold, the probability of lethal mutation occurring with each viral replication is so high that the viral population cannot grow, and hence, the population becomes extinct [16]. The error threshold is the mutation rate $u_e$ that solves $b(1-u_e)^m = a_0$.

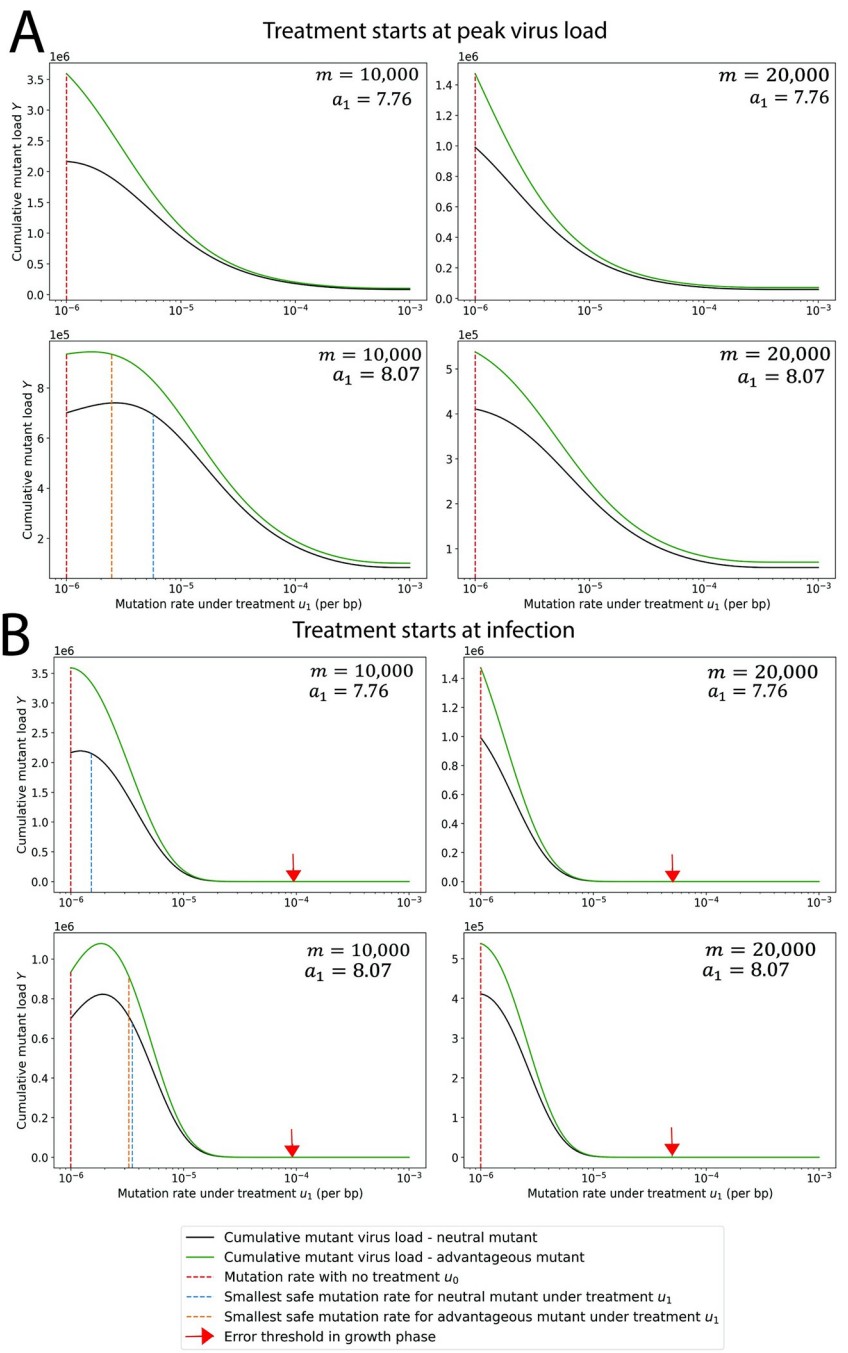

**Fig 2. Cumulative mutant virus load versus mutation rate, $u_1$, during treatment.** The cumulative mutant virus load increases with mutation rate $u_1$ before reaching a peak and then decreases to low values. If the peak is reached at a mutation rate that is less than the natural mutation rate, $u_0$ (red dotted line), then any increase in mutation rate reduces the cumulative mutant load. If the peak is reached for a mutation rate greater than $u_0$, then the increase in mutation rate caused by mutagenic treatment must exceed a threshold value (blue dotted line) to reduce the cumulative mutant virus load. We also consider mutants with a 1% advantage in the birth rate. As expected, we observe a higher cumulative mutant load for the advantageous mutant (green line) compared to the neutral mutant (blue line). But the minimum mutation rate under treatment that is required for evolutionary safety is slightly lower for the advantageous mutant. (A) Treatment starts at peak virus load. (B) Treatment starts at infection. The red arrow indicates the mutation rate at the error threshold of the growth phase. Parameters: $b = 7.61$ per day, $a_0 = 3$ per day, $n = 1$ position, $T = 5$ days, $m$ and $a_1$ as shown. The code used to generate this figure can be found at DOI: 10.5281/ zenodo.8017992.

### Treatment begins at (or soon after) infection

In **Fig 2**, we also show the cumulative mutant load, $Y(u_1)$, as a function of the mutation rate $u_1$ for the case where treatment starts at infection. We find that this function attains a maximum at a value which is given by the root of a third order polynomial (see **Methods** and **Fig A3** in **S1 Text**). Using the notation $k = [b(2b - a_0 - a_1)]/[(b - a_0)(a_1 - b)]$ and $h = bT$, we can approximate $u^*$ as follows:

$$\text{if } k > h \text{ then } u^* \approx 1/(km)$$

$$\text{if } k = h \text{ then } u^* \approx 0.52138/(hm)$$

$$\text{if } k < h \text{ then } u^* \approx 1/(hm). \tag{3}$$

Again if $u_0 > u^*$ then any increase in mutation rate is beneficial. If $u_0 < u^*$ then a small increase in the mutation rate can be evolutionarily not safe, but a sufficiently large increase in mutation rate can make the treatment evolutionarily safe (see **Fig 2** for more details). We also find that early treatment is more effective in reducing mutant virus load when compared to late treatment.

### Exploring the parameter space for evolutionary safety

In **Fig 3**, we show the fold increase in virus mutation rate that mutagenic treatment has to achieve beyond the innate mutation rate of the virus to be evolutionarily safe. We vary first the estimated number of lethal mutations $m$ in the viral genome and the clearance rate $a_1$. For treatment starting at peak virus load (**Fig 3A**), we find that increase in mutation rate is evolutionarily safe if $m > 22,000$ or $a_1 < 7.8$ per day (green region). Evolutionary safety becomes an issue for small values of $m$ and larger values of $a_1$. For $m = 12,000$ positions and $a_1 = 9$ per day, we need at least a 10-fold increase in mutation rate before the drug attains evolutionary safety. When treatment begins at infection (**Fig 3B**), the evolutionarily safe area becomes smaller, but the minimum increase in mutation rate required for evolutionary safety is lower. For example, for $a_1 = 9$ per day and $m = 12,000$ positions, we need only a 3-fold increase. We show the same figure, but for an extended range of $m$ values in **Fig A4** in **S1 Text**. Note that our estimate for fold increase in mutation rate for Molnupiravir is about 2, so the drug is safe only for a portion of the parameter space.

### Evolutionary risk factor (ERF) and infectiousness risk factor (IRF)

We define the "evolutionary risk factor" (ERF) of mutagenic treatment as the ratio of cumulative mutant virus load with treatment compared to without treatment (see **Methods**). The condition for evolutionary safety of mutagenic treatment is that ERF is less than one. Denote by $Y_{ij}$ the cumulative mutant load with the subscript $i$ indicating the presence ($i = 1$) or absence ($i = 0$) of treatment during the growth phase, and the subscript $j$ indicating the presence ($j = 1$) or absence ($j = 0$) of treatment during the clearance phase. Therefore, $Y_{00}$ is the cumulative mutant load without treatment, $Y_{01}$ is the cumulative mutant load with treatment in the clearance phase, and $Y_{11}$ is the cumulative mutant load with treatment in both growth and clearance phase. For treatment that starts at peak, $ERF = Y_{01}/Y_{00}$. For treatment that starts at infection, $ERF = Y_{11}/Y_{00}$. An evolutionary risk factor below one signifies that treatment reduces the mutant load, and hence, treatment can be even encouraged from an evolutionary perspective. An evolutionary risk factor above one implies that treatment increases the mutant load.

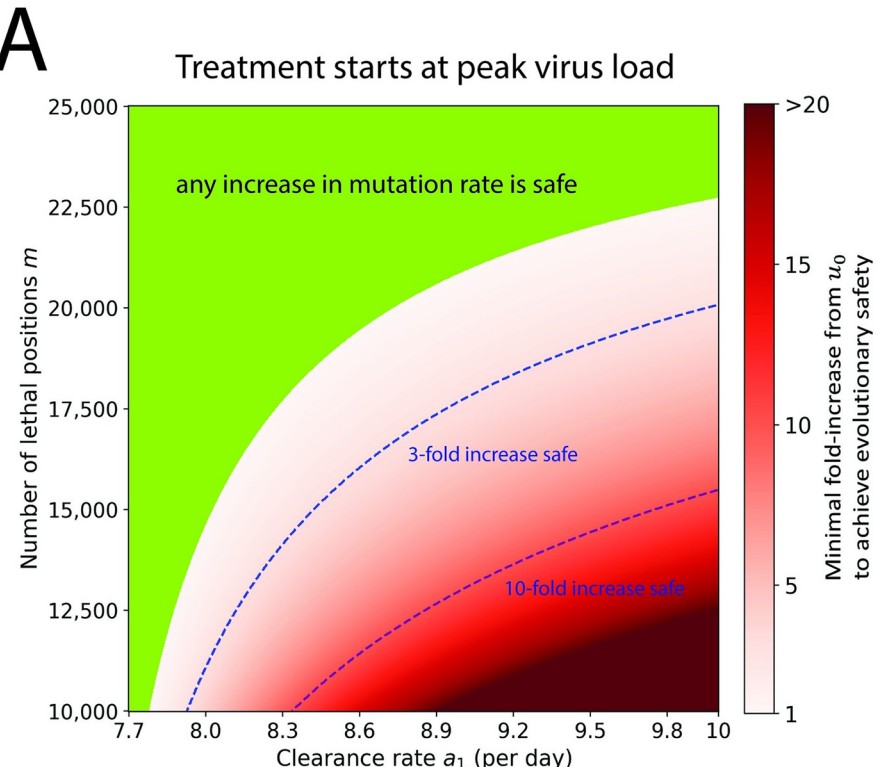

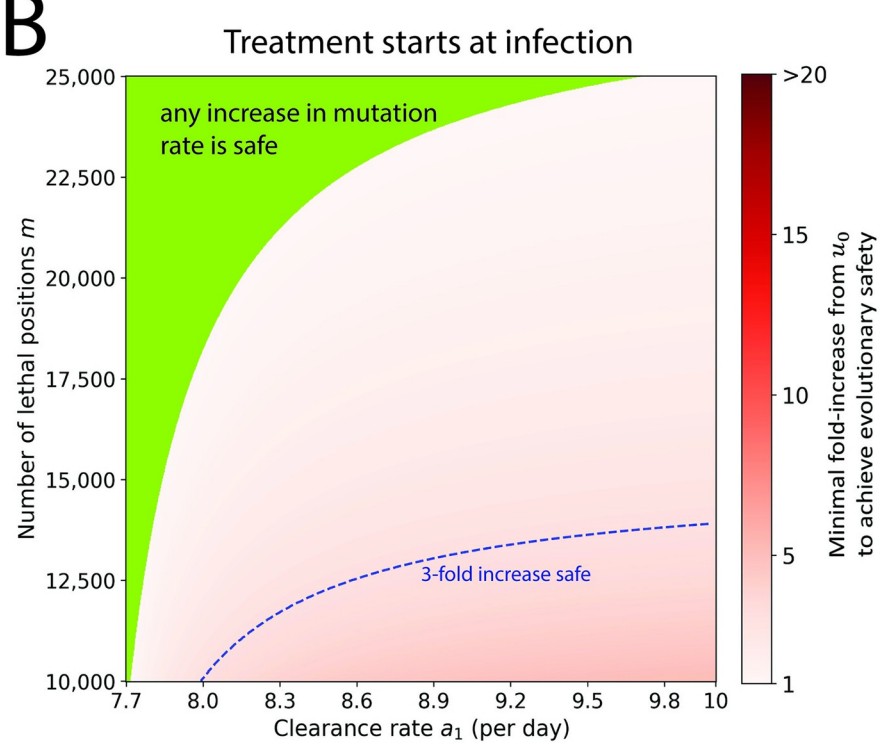

**Fig 3. Evolutionary safety of mutagenic treatment.** In the green parameter region, any increase in mutation rate reduces the cumulative mutant virus load and is therefore evolutionarily safe. In the red shaded region, we indicate the minimum fold increase in mutation rate that is required to reduce the cumulative mutant load. Contour lines for 3-fold and 10-fold increase are shown. (A) Treatment starts at peak virus load. (B) Treatment starts at infection. Parameters: $b = 7.61$ per day, $a_0 = 3$ per day, $n = 1$, $T = 5$ days, $u_0 = 10^{-6}$ per bp. The code used to generate this figure can be found at DOI: 10.5281/zenodo.8017992.

**Table 2. Cumulative virus load, mutant load, IRF, and ERF of mutagenic treatment.**

| Value of $a_1$ | Cumulative mutant viral load with no treatment $Y_{00}$ (×1,000) | Cumulative mutant viral load with treatment $Y_{01}$ (×1,000) | Evolutionary risk factor $Y_{01}/Y_{00}$ | Cumulative viral load with no treatment $V_{00}$ (×$10^9$) | Cumulative viral load with treatment $V_{01}$ (×$10^9$) | Infectivity risk factor $V_{01}/V_{00}$ |
|---|---|---|---|---|---|---|
| Treatment starts at peak virus load | | | | | | |
| 7.69 | 1,488 | 758 | **0.51** | 21.9 | 10.3 | **0.47** |
| 7.76 | 1,030 | 634 | **0.62** | 17.0 | 9.2 | **0.54** |
| 7.92 | 596 | 458 | **0.77** | 11.5 | 7.5 | **0.65** |
| 8.07 | 428 | 363 | **0.85** | 8.9 | 6.4 | **0.72** |
| 8.76 | 197 | 191 | **0.97** | 4.8 | 4.1 | **0.86** |
| Treatment starts at infection | | | | | | |
| 7.69 | 1,488 | 343 | **0.23** | 21.9 | 2.4 | **0.11** |
| 7.76 | 1,030 | 297 | **0.29** | 17.0 | 2.1 | **0.13** |
| 7.92 | 596 | 228 | **0.38** | 11.5 | 1.7 | **0.15** |
| 8.07 | 428 | 189 | **0.44** | 8.9 | 1.5 | **0.17** |
| 8.76 | 197 | 112 | **0.57** | 4.8 | 1.0 | **0.20** |

We show numerical results for individuals that differ in their immune competence, which affects the clearance rate, $a_1$, during adaptive immunity. Patients that are less immunocompetent benefit more from mutagenic treatment (lower IRF) and also have a lower ERF. Parameters: $a_0 = 3$ per day, $b = 7.61$ per day, $u_0 = 10^{-6}$ per bp, $u_1 = 3 \cdot 10^{-6}$ per bp, $m = 20,000$, $n = 1$, $T = 5$ days. Initial condition: $x_0 = 1$ and $y_0 = 0$.

ERF, evolutionary risk factor; IRF, infectiousness risk factor.

In addition, we define the "infectiousness risk factor" (IRF) that quantifies the efficacy of the treatment by killing and clearing the virus. The IRF is the ratio of the total cumulative viral load, mainly governed by the wild type, with treatment compared to the total cumulative viral load without treatment. IRF is always below 1.

In **Table 2**, we computed some values for the cumulative mutant and total virus load with and without treatment, as well as the corresponding ERF and IRF. We notice that ERF increases (hence evolutionary safety decreases) with immunological clearance rate, $a_1$. However, both the cumulative mutant viral load with and without treatment decrease with clearance rate. Hence, although the ERF is higher for more immunocompetent individuals, the absolute quantity of mutant produced is lower. We also notice that the IRF increases with immunocompetence, indicating that the benefit of treatment is smaller for more immunocompetent individuals who clear the virus rapidly even without treatment.

In **Fig 4** and **Fig A5** in **S1 Text**, we explore the ERF for wider regions of the parameter space. We vary each pair of parameters, while fixing others at their most probable value. The ERF exceeds 1 when the number of positions that would be lethal when mutated is much lower than our minimum estimate ($m < 12,000$). As $m$ decreases, treatment induces less lethal mutagenesis and thus provides more opportunity for mutants to be generated and to survive. Again, we observe that evolutionary safety decreases with the clearance rate, $a_1$. Delaying treatment, especially past the peak of the virus load, brings ERF closer to 1. Hence, early treatment for high enough $m$ should be encouraged since it can substantially decrease the abundance of mutant. Overall, we notice that most regions of the parameter space are evolutionarily safe. Since Molnupiravir's recommended course of treatment is only 5 days long, we also plotted the same figure for a 5-day treatment, starting at peak (see **Fig A6** in **S1 Text**). We observe no difference with treatment until virus clearance. When treatment is stopped, the virus load of viable mutant produced after the end of treatment is always much smaller than the virus load of viable mutant produced with no treatment for the same time frame (see **Fig A7** in **S1 Text**).

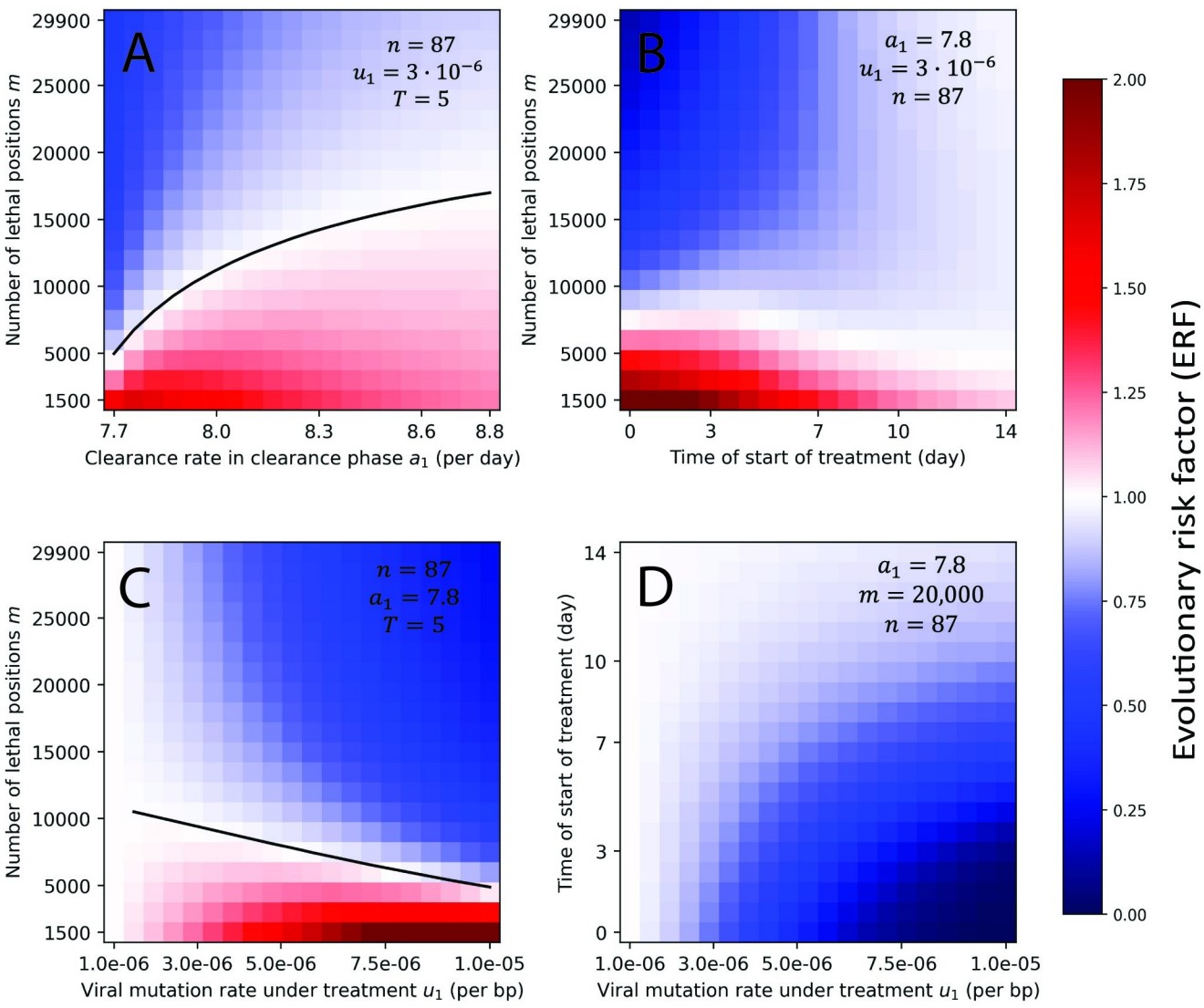

**Fig 4. ERF for a grid of parameters.** For each pair of parameters, we numerically compute the ERF for a range of values, while all other parameters are fixed. We observe that the value of $n$ has little effect on the ERF. Evolutionary risk factors above 1 are only observed for low values of the number of lethal positions, $m$. The ERF decreases with early treatment, high viral mutation rate under treatment, and large number of lethal positions. Initial condition: $x_0 = 1$ and $y_0 = 0$. The code used to generate this figure can be found at DOI: 10.5281/zenodo.8017992. ERF, evolutionary risk factor.

Hence, discontinuation of treatment after 5 days does not introduce an additional concern for evolutionary safety.

In **Fig A8** in **S1 Text**, we plot the results for $u_0 = 5 \cdot 10^{-6}$, and in **Fig A9** in **S1 Text**, we plot the results for $u_0 = 10^{-5}$. Considering $u_0 = 5 \cdot 10^{-6}$ results in greatly improved evolutionary safety. Mutagenic treatment remains unsafe only when $m = 1,500$, which is unrealistically small (see **Fig A8** in **S1 Text**). For $u_0 = 10^{-5}$, the treatment is safe for the entirety of the parameter space (see **Fig A9** in **S1 Text**). In **Fig A10** in **S1 Text**, we explore the ERF for lower and higher values of the birth rate $b$ and the clearance rate $a_0$ in the growth phase. We adjust the values of $b$ and $a_0$ such that the net growth rate is conserved (ignoring lethal mutations). We observe that smaller values of $b$ and $a_0$ lead to an increase in ERF, while larger values to a decrease. Variants of SARS-CoV-2, such as Delta or Omicron, have exhibited differing times

to the peak of virus load and the value of the virus load at peak. We included a sensitivity analysis on these 2 parameters in **Fig A11** in **S1 Text**.

In order to study the effect of genetic drift on the ERF, we implemented a stochastic version of our model using the Gillespie algorithm with tau-leaping [69]. We found that incorporating genetic drift into the calculation results in increased evolutionary safety of a treatment for parameter sets where the ERF was higher than 1 (see **Fig A12** in **S1 Text**).

## The evolutionary risk factor is a slowly declining function of the number of mutations leading to viable virus

So far, we have used the parameter $n = 87$ to denote the number of mutations that would result in VoCs that is variants with increased transmissibility, virulence, or resistance to existing vaccines and treatments. However, in the broad sense, any treatment that increases the standing genetic variation of the virus could favor the emergence of new variants of concern by enabling epistatic mutations. Therefore, we now extend the interpretation of $n$ to include any viable mutation in the viral genome.

In **Fig 5**, we show that the ERF is a declining function of $n$. Thus, the more opportunities the virus has for viable mutations (the larger $n$), the higher the advantage of mutagenic treatment. The reason for this counter-intuitive observation is that for large $n$ the cumulative mutant virus load is high already in the absence of treatment, while mutagenic treatment reduces the mutant load by forcing additional lethal mutations. ERF decreases with the number of positions $n$ also for lower birth rate $b$ (**Fig A13** in **S1 Text**).

## Advantageous mutants do not substantially affect the evolutionary safety compared to neutral mutants

Mutants could have an in-host advantage compared to wild type, such as faster a reproductive rate or a lower clearance rate. In **Fig 2**, we evaluate a mutant with a 1% selective advantage in birth rate. We have also included results for a mutant with a 0.5% selective advantage in **Fig A14** in **S1 Text**, and results for mutants with 0.5% and 1% selective disadvantage in **Figs A15** and **A16** in **S1 Text.** As expected, we observe that the advantageous mutant reaches higher virus load than a neutral mutant. But we also observe that if there is a minimum increase in mutation rate that is required for evolutionary safety, then it is lower (or slightly lower) for the advantageous mutant. Therefore, a treatment that is evolutionarily safe for a neutral mutant is also evolutionarily safe for an advantageous mutant. Conversely, we observe that the minimum increase in the mutation rate required for evolutionary safety is higher when the mutant has a selective disadvantage.

We also consider a 1% disadvantage of the wild type with regards to the mutant that is exhibited under treatment, that is when $u_0 > 10^{-6}$. In this scenario, we find that the treatment is always evolutionarily safe (see **Fig A17** in **S1 Text**).

## Gradual activation of the immune system

So far, we have considered a sudden activation of the adaptive immune response by switching the clearance from $a_0$ to $a_1$ at time $T$ resulting in a two-phase model of immunity. In reality, the immune response intensifies gradually over the course of the infection [16]. We explore a more gradual onset of the immune response in **Fig A18** in **S1 Text**, where we add an intermediate phase during which the clearance rate is the arithmetic average of $a_0$ and $a_1$. We find that the ERF value for the three-phase immunity is very close to and bounded by the ERF values found for corresponding two-phase simulations.

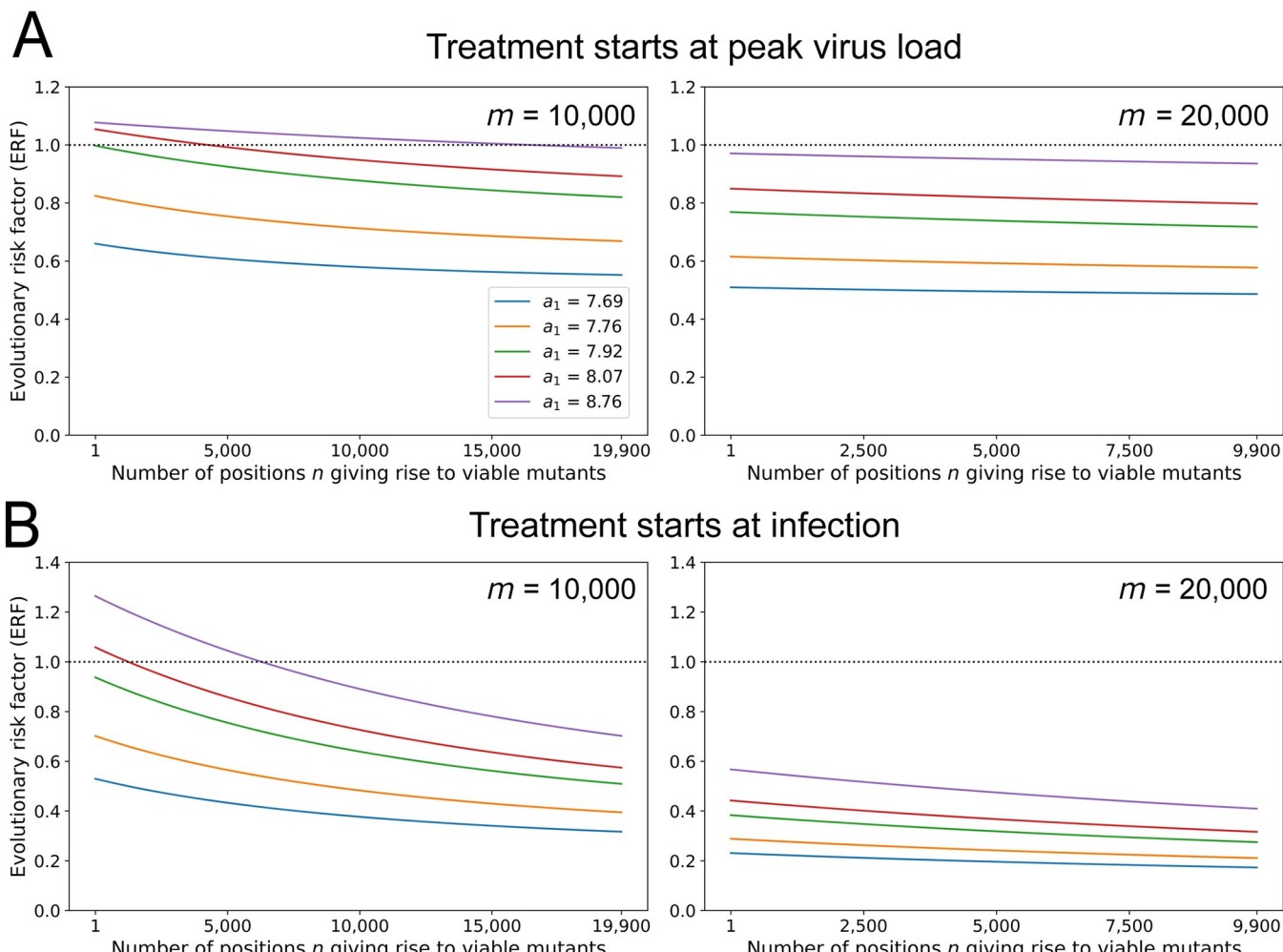

**Fig 5. The ERF versus the number, *n*, of positions in the viral genome giving rise to concerning (or viable) mutations.** The ERF of mutagenic treatment is the ratio of the cumulative mutant virus load with and without treatment. We explore all values of *n* subject to the constraint that *m+n* remains below the length of the SARS-CoV-2 genome. We observe that the ERF decreases as function of *n*. (A) Treatment starts at peak virus load. (B) Treatment starts at infection. Parameters: $a_0 = 3$ per day, $b = 7.61$ per day, $u_0 = 10^{-6}$ per bp, $u_1 = 3 \cdot 10^{-6}$ per bp, $T = 5$ days. Initial condition: $x_0 = 1$ and $y_0 = 0$. The code used to generate this figure can be found at DOI: 10.5281/zenodo.8017992. ERF, evolutionary risk factor; SARS-CoV-2, Severe Acute Respiratory Syndrome Coronavirus 2.

## Nonlethal deleterious mutations

In order to make the model more biologically realistic, we have extended our model to consider nonlethal deleterious mutations. In our extended model, we consider 4 categories of virus: the wild-type; nonlethal deleterious mutants with no concerning mutations; mutants with concerning mutations; and nonlethal deleterious mutant with concerning mutations. We found that considering nonlethal deleterious mutations always increases the evolutionary safety of the treatment. For a detailed analysis, see section "Non-lethal deleterious mutants" in S3 Text.

## Lethal defection

Nonviable virions can negatively interfere with the replication of viable virions that are being generated from the same cell. For example, nonviable virions may consume functional

proteins that are then lacking for the replication and packaging of the viable virions. Conversely, nonfunctional proteins synthetized from the mutant genomes can lead to the production of noninfective virus particles containing wild-type genomes.

We extended our model to take into account the interference of nonviable mutants in the replication of viable mutants. We found that the evolutionary safety is always increased when incorporating this effect. For a detailed analysis, see S3 **Text**.

## A simple approach captures the essence of mutagenic treatment and evolutionary safety

We further simplify our mathematical framework to obtain quantitative guidelines about the evolutionary safety of a mutagenic drug. We find that focusing on virus dynamics in the growth phase can be used to approximate the full infection dynamics, especially if the clearance rate is large. Note that clearance rates leading to infections which last longer than 100 days remain exceptions, and hence, most individuals have a high clearance rate $a_1$. The simplified approach is presented in the **Methods**. The agreement between the simplified and the full model is shown in **Fig A19** in **S1 Text**.

The eventual goal of all mutagenic treatments would be to prevent the exponential expansion of the virus even before the onset of adaptive immunity. Using the SARS-CoV-2 estimates, $m = 20,000$ positions, $b = 7.61$ per day, and $a_0 = 3$ per day, we find that mutagenic treatment would have to achieve $u_1 > 4.65 \cdot 10^{-5}$, which is a 50-fold increase of the natural mutation rate of the virus. If the mutagenic drug results in a smaller increase in the virus mutation rate under treatment, then it not prevent the establishment of the infection, but it could still reduce both wild type and mutant abundance. The mutant virus load at time $T$ is a one-humped function of the mutation rate with a maximum that is close to $u^* = 1/(bTm)$. For $m = 20,000$ positions, $b = 7.61$ per day, and $T = 5$ days, we find $u^* = 1.32 \cdot 10^{-6}$ per bp. This value is close to the estimate of the natural mutation rate of the virus, $u_0 = 10^{-6}$ per bp. If $u_0$ was greater than $u^*$ then any increase in mutation rate would be evolutionarily safe. Otherwise, we need to calculate the condition for evolutionary safety. Let us introduce the parameter $s$ with $u_1 = su_0$. The condition for evolutionary safety in the simplified model is

$$m > \frac{\log s}{bTu_0(s-1)}. \tag{4}$$

As before $b = 7.61$ per day, $T = 5$ days, and $u_0 = 10^{-6}$ per bp. For $s = 3$ fold increase of mutation rate induced by mutagenic treatment, we get $m > 14,455$ positions. Since evolutionary safety improves with decreasing clearance rate $a_1$ (in the full model), we can interpret inequality (4) as a sufficient condition or as an upper bound. The agreement between the analytical formulas and the numerical computation of the model is shown in **Fig A20** in **S1 Text**. For the simplified model, we also find that ERF is a declining function of the number of viable mutations, $n$ (see **Fig A21** in **S1 Text**).

## Evolutionary safety for higher-order mutants

Up until now, we considered only one-step mutations in order to produce viable virus. However, many examples of multistep adaptation have been observed in virus evolution. In particular, the H274Y mutation has been studied and found to be deleterious, but the loss in fitness could be restored by other mutations [61,62]. In this section, we consider double mutants. The

evolutionary dynamics for a virus with a two-locus, binary genome can be written as:

$$\dot{y}_{00} = B_{00}q^2 y_{00} + B_{01}q(1-q)y_{01} + B_{10}q(1-q)y_{10} + B_{11}(1-q)^2 y_{11} - a_j y_{00}$$

$$\dot{y}_{01} = B_{00}q(1-q)y_{00} + B_{01}q^2 y_{01} + B_{10}(1-q)^2 y_{10} + B_{11}q(1-q)y_{11} - a_j y_{01}$$

$$\dot{y}_{10} = B_{00}q(1-q)y_{00} + B_{01}(1-q)^2 y_{01} + B_{10}q^2 y_{10} + B_{11}q(1-q)y_{11} - a_1 y_{10}$$

$$\dot{y}_{11} = B_{00}(1-q)^2 y_{00} + B_{01}q(1-q)y_{01} + B_{10}q(1-q)y_{10} + B_{11}q^2 y_{11} - a_j y_{11}. \tag{5}$$

The frequencies $y_{00}$, $y_{01}$, $y_{10}$, and $y_{11}$ denote, respectively, the wild-type virus (00), 2 single mutants (01 and 10), and the double mutant (11). We have $q = 1-u$, where $q$ is the quality of replication, and $u$ is the mutation rate.

We have $B_{ij} = (1+ds_{ij})b$, where $B_{ij}$ the birth rate of the variant $ij$, $d$ is the fitness difference between the wild type and the mutant, and $s_{ij}$ determines whether the mutant has a fitness disadvantage ($s_{ij} = -1$), is neutral ($s_{ij} = 0$) or has a fitness advantage ($s_{ij} = 1$). For the wild type (00), we always set $s_{00} = 0$, hence $B_{00} = b$.

The fitness landscape is given by the vector ($s_{01}$, $s_{10}$, $s_{11}$). Since each component has one of 3 values (−1, 0, or 1), there are 27 possible landscapes. Because of symmetry, 9 landscapes are redundant. Hence, we are left with 18 different landscapes.

When considering double mutants, we propose 2 different methods for evaluating evolutionary safety.

In Method 1, evolutionary safety requires that treatment reduces the sum of all mutants (single and double). In this case, double mutants make only a negligible contribution to evolutionary safety because their abundance is much lower than that of single mutants (see **Figs A22–A24** in **S1 Text**).

In Method 2, evolutionary safety requires that treatment reduces the amount of each mutant type separately. Here, we find that in some cases evolutionary safety requires a larger increase in mutation rate in order to reduce the amount of double mutant (see **Figs A25–A27** in **S1 Text**).

In **Figs A28** and **A29** in **S1 Text** as well as in Tables **A3** and **A4** in **S1 Text**, we investigate the effect of different fitness landscapes on evolutionary safety. In **Fig A26** in **S1 Text** and in **Table A3** in **S1 Text**, we consider a patient that has a low clearance rate of infection. In **Fig A29** in **S1 Text** and in **Table A4** in **S1 Text**, we consider a patient that has a high clearance rate of infection (where in general mutagenic treatment is less likely to be evolutionarily safe).

In **Fig A28** in **S1 Text**, we see that any increase in mutation rate reduces the cumulative sum of all mutants and is therefore evolutionarily safe using Method 1. For 4 of the 5 landscapes shown here, any increase in mutation rate reduces the abundance of the double mutant and is therefore evolutionary safe using Method 2. In **Fig A29** in **S1 Text**, we see that any increase in mutation rate reduces the cumulative sum of all mutants and is therefore evolutionarily safe using Method 1. For all 5 landscapes shown here, we need roughly a 3-fold increase in mutation rate to reduce the abundance of the double mutant and therefore achieve evolutionary safe using Method 2.

In **Table A3** in **S1 Text**, we see that for 14 of the 18 fitness landscapes, any increase in mutation rate reduces the abundance of the double mutant, but for 4 landscapes we need an increase in mutation rate between 1.2- and 1.5-fold to reduce the abundance of the double mutant. In **Table A4** in **S1 Text**, we see that for all 18 fitness landscapes, we need

approximately a 3.5- to 3.8-fold increase in mutation rate to reduce the abundance of the double mutant.

## Weighted evolutionary safety

One can also define evolutionary safety for specific mutants that have been identified as potentially dangerous. Hence, their cumulative amount produced with and without treatment can be weighted by a higher factor than other, less dangerous mutants (see section in **Methods**: "Weighted ERF"). We explore this extension of the ERF in **Fig A30** in **S1 Text**. We find that associating different weights to different mutants has no effect on the ERF. This is because the different mutants differ only by the number of positions that give rise to them when mutated, which has a very limited effect on the ERF (see **Fig 5**).

## Discussion

We provide a mathematical framework to compute the evolutionary risk factor of death caused by mutagenic drugs and apply it to Molnupiravir, SARS-CoV-2, and COVID-19. We define evolutionary safety as the situation in which the cumulative virus load with treatment is less or equal to the cumulative virus load without treatment.

For our current estimates of the parameter space, Molnupiravir treatment appears to be evolutionarily safe and can be encouraged for individuals with low clearance rates. For individuals with high clearance rates, the treatment might increase the rate of emergence of new mutants by a few percent. However, the excess of mutant produced by individuals with fast immunological clearance upon treatment is small in absolute amount due to the relatively smaller cumulative mutant virus load generated in such individuals. Treatment of individuals with low clearance rate tends to be evolutionarily safe, since it greatly reduces the amount of virus, and thus, potential mutants, compared to no treatment. Note that in this paper, we have adopted a stringent requirement for evolutionary safety, namely that the quantity of all generated mutants with treatment be lower or equal than without treatment, be they actual VoCs or any other mutant. Extending our model to consider drift, nonlethal deleterious mutants and lethal defection has only increased evolutionary safety of treatment compared to lack of treatment. Yet, our conclusions are still contingent on parameter estimation correctness, and as show above, a change in estimation of some critical parameter might render the treatment non-safe.

Virus kinetics models have been used extensively to inform antiviral treatment, also in the context of the COVID-19 pandemic. For example, Kern and colleagues have suggested a method to guide drug repurposing and estimate the optimal time window for SARS-CoV-2 treatment [70]. Modeling virus kinetics can also provide insights into SARS-CoV-2 pathogenesis [71]. Virus kinetics models are also useful in order to estimate basic data about SARS-CoV-2 infections, such as the incubation time, time of viral shedding, and clearance time [49,50,72–76].

Although the use of mutagenic treatments has caused some popular concern on social media, no one has so far, to our knowledge, attempted to assess the evolutionary safety of mutagenic drugs through rigorous mathematical modeling.

Mutagenic treatment acts to decrease the total virus load by causing lethal mutations. It can also decrease the mutant load since (i) it eliminates the ancestors of viable mutants; and (ii) it accelerates the demise of their offspring by inducing lethal mutations. In some individuals with slow clearance rates, for whom the cumulative virus load without treatment is high, mutagenic treatment can substantially reduce the amount of mutant virus generated over the course of an infection. In immunocompetent individuals, the positive effect of mutagenic treatment

on reducing virus load is smaller and the abundance of mutant virus can even be increased. A graphical summary of this intuition is shown in **Fig 1C**.

The main limitation of our study is the lack of transition from the cumulative mutant load to risk of spreading in the population. Although some studies attempted to link epidemiology with the infectiousness along time of individuals [77–80], expanding our model to the epidemiological assessment of VoCs generated by mutagenic treatments is beyond the scope of our study. Furthermore, our knowledge about SARS-CoV-2 is still evolving. Hence, estimates for key parameters, such as the number of positions that are lethal when mutated, could change. If new estimates were to show that the value of $m$ is below 12,000, then we predict that the evolutionary risk factor of Molnupiravir exceeds 1, and hence, the treatment could increase the rate of appearance of new potentially concerning mutants. We therefore advocate caution when drawing conclusions about Molnupiravir's safety. However, our analysis has also identified parameters which will not affect appreciably the assessment of evolutionary safety of Molnupiravir, such as the number of positions that are able to give rise to viable mutants. In addition, if it turns out that early antiviral treatment delays the onset of the immune response, then evolutionary safety of early treatment would be considerably reduced.

Our analysis has also provide a simple rule (Eq 4) for evolutionary safety of mutagenic treatment. We anticipate that additional lethal mutagenesis drugs will emerge, and their evolutionary safety will need to be assessed before making them available for treatment. For instance, Favipiravir has been suggested as another mutagenic treatment for SARS-CoV-2 [81].

The safety concerns that emerge from the use of a mutagenic drug extend beyond the increased rate of appearance of new VoCs. Additional deleterious effects of Molnupiravir may include the mutagenesis of the host DNA following metabolic conversion of the drug into $2'$-deoxyribonucleotide [14] and putative toxic effects on transcription of the host RNA. In addition, mutagenic treatment can have off-target effects in the event of coinfection with several pathogens. These other toxic effects are outside the scope of the current study.

Finally, the framework presented here is general enough for the assessment of evolutionary safety of this and other mutagenic drugs, in the treatment of other infectious diseases and their pathogens. Our analytical and simulation code is available on-line for further explorations (see Data availability statement).

## Methods

We denote by $x$ and $y$ the abundances of wild-type and mutant virus in an infected person. Evolutionary dynamics can be written as follows:

$$\dot{x} = x(bq^{m+n} - a_j) \tag{5A}$$

$$\dot{y} = xbq^m(1 - q^n) + y(bq^m - a_j). \tag{5B}$$

The parameter $b$ denotes the birth (or replication) rate of the virus. The parameter $a_j$ denotes the death (or clearance) rate of the virus. The subscript $j$ indicates the absence ($j = 0$) or presence ($j = 1$) of an adaptive immune response. We have $a_1 > b > a_0$. The accuracy of viral replication is given by $q = 1 - u$, where $u$ is the virus mutation rate per base. The number of lethal (or highly deleterious) positions in the viral genome is given by $m$. The number of positions in the viral genome leading to viable mutants is given by $n$. Therefore, $y$ measures the abundance of mutants in a patient. At first, we assume that those mutations are neutral in the sense of having the same parameters $b$ and $a_j$ as the wild-type virus in the patient in which they arise. We note that in Eq (5), the mutant is mildly advantageous because $q^m > q^{m+n}$. We assume that the adaptive immune response begins $T$ days after infection, at which time the clearance

rate of the virus increases from $a_0$ to $a_1$. Therefore, peak virus load is reached at time $T$. For exponential increase in virus load during the growth phase, which occurs during the first $T$ days of infection, we require $bq_0^{m+n} > a_0$. For exponential decrease in virus load during the clearance phase, we require $bq_0^m < a_1$.

Using $v = x+y$ for the total virus abundance, we obtain

$$\dot{v} = v(bq^m - a_j). \tag{6}$$

Eq (6) is the same as Eq (5A), but $m$ occurs instead of $m+n$. In the following, we derive results for $v$. The corresponding results for $x$ are obtained by replacing $m$ with $m+n$. Results for $y$ are given by $v-x$. During the growth phase, we have $\dot{v} = v(bq^m - a_0)$. For initial condition $v = 1$ we get

$$v(t) = e^{(bq^m - a_0)t}. \tag{7}$$

The cumulative amount of virus produced until time $T$ is

$$V^+ = \int_0^T v(t)dt \approx \frac{1}{bq^m - a_0} e^{(bq^m - a_0)T}. \tag{8}$$

Neglecting the term $1/(bq^m - a_0)$. The growth phase ends at time $T$, at which point the virus abundance is

$$v_T = e^{(bq^m - a_0)T}. \tag{9}$$

We use $v_T$ and the corresponding quantities $x_T$ and $y_T$ as initial conditions for the clearance phase. For the clearance phase, which starts at time $T$, we have $\dot{v} = -v(a_1 - bq^m)$. Using initial condition $v_T$, we obtain

$$v(t) = v_T e^{-(a_1 - bq^m)t}. \tag{10}$$

The cumulative virus during the clearance phase is given by

$$V^- = \int_0^\infty v(t)dt = \frac{v_T}{a_1 - bq^m} = \frac{1}{a_1 - bq^m} e^{(bq^m - a_0)T}. \tag{11}$$

For the cumulative virus load of growth plus clearance phase, we obtain

$$V = V^+ + V^- \approx \left( \frac{1}{bq^m - a_0} + \frac{1}{a_1 - bq^m} \right) e^{(bq^m - a_0)T}. \tag{12}$$

Let us use $V_{ij}$ to denote the cumulative virus during the entire infection, where $i = 0$ or $i = 1$ indicates absence or presence of treatment during the growth phase and $j = 0$ or $j = 1$ indicates absence or presence of treatment during the clearance phase. We have

$$V_{ij} \approx \left( \frac{1}{bq_i^m - a_0} + \frac{1}{a_1 - bq_j^m} \right) e^{(bq_i^m - a_0)T}. \tag{13}$$

The corresponding equation for the cumulative wild-type virus is

$$X_{ij} \approx \left( \frac{1}{bq_i^{m+n} - a_0} + \frac{1}{a_1 - bq_j^{m+n}} \right) e^{(bq_i^{m+n} - a_0)T}. \tag{14}$$

The corresponding equation for the cumulative mutant virus is given by the difference

$$Y_{ij} = V_{ij} - X_{ij}. \tag{15}$$

Without any treatment, the cumulative mutant virus is $Y_{00}$. If treatment starts at time $T$, the cumulative mutant virus is $Y_{01}$. If treatment starts at time 0, the cumulative mutant virus is $Y_{11}$. Mutagenic treatment increases the mutation rate of the virus from $u_0$ to $u_1$ and therefore reduces the replication accuracy from $q_0$ to $q_1$. We have $u_0 < u_1$ and $q_0 > q_1$.

## Evolutionary risk factor

We define the evolutionary risk factor, *ERF*, of mutagenic treatment as the ratio of cumulative mutant virus load with treatment over the cumulative mutant virus load without treatment. For treatment that starts at time $T$, we have $ERF = Y_{01}/Y_{00}$. For treatment that starts at time 0, we have $ERF = Y_{11}/Y_{00}$. The *ERF* quantifies how safe or unsafe a mutagenic treatment is. If $ERF < 1$ then the treatment is evolutionarily safe.

## Infectiousness risk factor

We define the infectiousness risk factor, *IRF*, of mutagenic treatment as the ratio of cumulative virus load with treatment over the cumulative virus load without treatment. For treatment that starts at time $T$, we have $IRF = V_{01}/V_{00}$. For treatment that starts at time 0, we have $IRF = V_{11}/V_{00}$.

## Treatment starts at peak virus load, *t = T*

The cumulative virus during the clearance phase with treatment is

$$V^- = \frac{v_T}{a_1 - bq_1^m}. \tag{16}$$

The cumulative wild-type virus during clearance phase with treatment is

$$X^- = \frac{x_T}{a_1 - bq_1^{m+n}}. \tag{17}$$

The cumulative mutant virus during clearance phase with treatment is

$$Y^- = V^- - X^- = \frac{v_T}{a_1 - bq_1^m} - \frac{x_T}{a_1 - bq_1^{m+n}}. \tag{18}$$

Using $v_T = x_T + y_T$, we write

$$Y^- = \frac{x_T + y_T}{a_1 - bq_1^m} - \frac{x_T}{a_1 - bq_1^{m+n}}. \tag{19}$$

Introducing $\eta = y_T/x_T$, we write

$$Y^- = x_T \left[ \frac{1 + \eta}{a_1 - bq_1^m} - \frac{1}{a_1 - bq_1^{m+n}} \right]. \tag{20}$$

From above, we have $x_T = \exp[(bq_0^{m+n} - a_0)T]$ and $v_T = \exp[(bq_0^m - a_0)T]$, which in turn specify $y_T$ and $\eta$. For the parameters that are relevant to us, we find that $Y^-$ as a function of the mutation rate $u_1$ that is induced during treatment has the following behavior (see **Fig A31** in **S1 Text**):

1. If $\eta \gg n/m$, then $Y^-(u_1)$ is a declining function. In this case, mutagenic treatment is always beneficial.

2. If $\eta \ll n/m$, then $Y^-(u_1)$ has a single maximum which is attained at

$$u^* = \frac{a_1 - b}{mb} \frac{n - \eta m}{n + \eta m}. \tag{21}$$

If $u_0 > u^*$, then any mutagenesis treatment is beneficial. If $u_0 < u^*$, then mutagenic treatment needs to be sufficiently strong to be beneficial; specifically, we need $Y^-(u_0) > Y^-(u_1)$, where $u_1 > u_0$. For small $u_0$, the condition $\eta > n/m$ is equivalent to $bT > 1/[mu_0(1 - mu_0)]$.

## Treatment starts at infection, $t = 0$

For relevant parameters, the cumulative mutant virus load $Y_{11}(u_1)$—given by Eq (15)—as a function of the mutation rate during treatment attains a single maximum at a value $u^*$. If $u_0 > u^*$, then mutagenic treatment is always beneficial. If $u_0 < u^*$, then mutagenic treatment needs to result in a sufficient increase in the virus mutation rate to be beneficial; specifically, we need $Y_{11}(u_0) > Y_{11}(u_1)$. We obtain $u^*$ as follows. Let $\mu = mu$. We find $\mu^* = mu^*$ as the solution of the polynomial:

$$F(\mu) = h + k - \mu(h^2 + k^2) - \mu^2 hk(2h + k) - \mu^3 h^2 k^2. \tag{22}$$

Here, $h = bT$ and $k = [b(2b - a_0 - a_1)]/[(b - a_0)(a_1 - b)]$. Exact solutions can be obtained but include complicated expressions. Approximate solutions can be found as follows. Consider fixed $h$ and declining $k$. As $k$ declines $\mu^*$ increases. There are 5 regions:

1. If $k \gg h$, then $\mu^* = 1/k$

2. If $k = h$, then $\mu^* = 0.52138/k = 0.52138/h$

3. If $h > k > 0$, then $\mu^* < 1/h$

4. If $h > k = 0$, then $\mu^* = 1/h$

5. If $h > 0 > k$, then $\mu^* > 1/h$ (but $\mu^*$ stays close to $1/h$).

Therefore, one can approximate as follows:

1. If $k > h$, then $\mu^* \approx 1/k$

2. If $k > h$, then $\mu^* \approx 0.52138/h$

3. If $k < h$, then $\mu^* \approx 1/h$.

See **Fig A3** in **S1 Text** for validity of those approximations. The full derivation of Eq 22 is provided in S2 **Text.**

## Evolutionary safety in a simplified setting

We now consider the effect of mutagenic treatment in a setting that uses further simplification. We only study the amount of virus that is generated during the growth phase with and without mutagenic treatment. As before we have:

$$\dot{x} = x(bq^{m+n} - a) \tag{23A}$$

$$\dot{y} = xbq^m(1 - q^n) + y(bq^m - a). \tag{23B}$$

For the total virus, $v = x+y$, we have:

$$\dot{v} = v(bq^m - a). \tag{23C}$$

We use $q = q_0 = 1 - u_0$ to denote absence of treatment and $q = q_1 = 1 - u_1$ to denote presence of treatment, with $u_1 > u_0$. In the absence of treatment, we assume $bq_0^{m+n} > a$, which means the wild type can expand.

Clearly, the aim of mutagenic treatment is to eradicate the infection, which is to prevent the exponential expansion. Thus, mutagenic treatment succeeds if $bq_1^m < a$. In other words, the mutation rate induced by mutagenic treatment should satisfy

$$u_1 > \frac{\log(b/a)}{m}. \tag{24}$$

Using our SARS-CoV-2 estimates, $m = 20{,}000$ positions, $b = 7.6$ per day, and $a = 3$ per day, we obtain $u_1 > 4.65 \cdot 10^{-5}$ per bp. If the natural mutation rate is $10^{-6}$ per bp then—ideally—we are looking for a mutagenic drug that achieves a 50-fold increase in mutation rate.

If the mutagenic drug induces a smaller fold increase in virus mutation rate, then it does not prevent the infection, but it could still reduce both virus load and mutant virus load. In this case, a more complicated calculation is needed. For initial condition $v = 1$ ($x = 1$ and $y = 0$), we obtain at time $T$

$$v(T) = e^{(bq^m - a)T} \tag{25A}$$

$$x(T) = e^{(bq^{m+n} - a)T} \tag{25B}$$

$$y(T) = e^{(bq^m - a)T} - e^{(bq^{m+n} - a)T}. \tag{25C}$$

We need to understand how $y_T$ behaves as a function of the mutation rate. For this analysis, the parameter $a$ is irrelevant, because we can write

$$y(T) = e^{-aT}\left(e^{bTq^m} - e^{bq^{m+n}T}\right). \tag{26}$$

We find that $y_T(u)$ is a one-humped function with a single maximum near

$$u^* = \frac{1}{bTm}. \tag{27}$$

This approximation holds for $mu^* \ll 1$. Increasing $b$, $T$, or $m$ reduces the value of $u^*$. If $u_0$ is greater $u^*$, then any increase mutation rate reduces the amount of mutant virus. Using our SARS-CoV-2 estimates, $m = 20{,}000$ positions, $b = 7.6$ per day, and $T = 5$ days, we obtain $u^* = 1.31 \cdot 10^{-6}$ per bp. This value is very close to the estimate for the normal mutation rate $u_0 = 10^{-6}$. If $u_0$ is less than $u^*$, then we need to calculate the ERF to evaluate if the treatment reduces the amount of mutant virus. We have

$$ERF = \frac{e^{(bq_1^m - a)T} - e^{(bq_1^{m+n} - a)T}}{e^{(bq_0^m - a)T} - e^{(bq_0^{m+n} - a)T}}. \tag{28}$$

Notice that $a$ cancels out and the parameters $b$ and $T$ appear as the product $h = bT$. We obtain

$$ERF = \frac{e^{hq_1^m} - e^{hq_1^{m+n}}}{e^{hq_0^m} - e^{hq_0^{m+n}}}. \tag{29}$$

Using the approximation $q^{m+n} = (1-u)^{m+n} \approx 1 - u(m+n)$, we get

$$ERF = \frac{e^{-hmu_1}\left(1 - e^{hnu_1}\right)}{e^{-hmu_0}\left(1 - e^{hnu_0}\right)}. \tag{30}$$

For small $hnu$, we can approximate $e^{-hnu} \approx 1 - hnu$, and therefore

$$ERF = \frac{u_1 e^{-hmu_1}}{u_0 e^{-hmu_0}}. \tag{31}$$

We find $ERF < 1$ if

$$u_1 e^{-hmu_1} < u_0 e^{-hmu_0}. \tag{32}$$

Which means

$$m > \frac{\log s}{hu_0(s-1)}. \tag{33}$$

The key parameter, $h = bT$, is the number of replication events between the infecting virion and those virions that are present at the time of evaluation; using $b = 7.61$ per day and $T = 5$ days, we have $h = 38.05$. For $u_0 = 10^{-6}$ per bp and $s = 3$ fold increase induced by mutagenic treatment, we get $m > 14{,}455$. For $s = 2$, we get $m > 18{,}217$.

Defining the infectiousness risk factor, $IRF$, as $v_1(T)/v_0(T)$, we obtain

$$IRF = \frac{e^{(bq_1^m - a)T}}{e^{(bq_0^m - a)T}} = \frac{e^{hq_1^m}}{e^{hq_0^m}} = e^{h(q_1^m - q_0^m)}. \tag{34}$$

Using the approximation $q^m = (1-u)^m \approx 1 - mu$, which holds for $u \ll 1$ we have

$$IRF = e^{-hm(u_1 - u_0)}. \tag{35}$$

We note that IRF is always less than 1.

## Treatment increases the mutation rate only in a fraction $f$ of positions

Molnupiravir is molecularly similar to a cytosine; however, it can base-pair equally efficiently with both adenosine and guanosine. Hence, the probability of certain possible mutations will be increased more than others. Specifically, in the case of Molnupiravir, transition mutations will be more frequent, but transversion mutations are not expected to increase. If the mutagenic drug increases the mutation rate in a fraction $f$ of positions, evolutionary dynamics can be written as

$$\dot{x} = x(bq_0^{(m+n)(1-f)}q_1^{(m+n)f} - a_j)$$

$$\dot{y} = xbq_0^{m(1-f)}q_1^{mf}(1 - q_0^{n(1-f)}q_1^{nf}) + y(bq_0^{m(1-f)}q_1^{mf} - a_j). \tag{36}$$

Let $q_2 = q_0^{1-f}q_1^f$. Hence, we have:

$$\dot{x} = x(bq_2^{m+n} - a_j)$$

$$\dot{y} = xbq_2^m(1 - q_2^n) + y(bq_2^m - a_j), \tag{37}$$

which is equivalent to Eq 5. Hence, all the subsequent derivations hold.

## Fraction of lethal mutations

Let us consider a genome of length $L$. Each position can mutate to 3 other nucleotides, hence, we have $M = 3L$, where $M$ is the total number of all possible mutations. The mutation rate per position is $u$. Let us assume that a proportion $p$ of the $M$ possible mutations is lethal. Hence, the probability of not acquiring a lethal mutation during the replication of the genome is $(1-u/3)^{pM}$. We have:

$$(1 - u/3)^{pM} = \exp\left(\log\left(\left(1 - \frac{u}{3}\right)^{pM}\right)\right) = \exp\left(pM \, log\left(1 - \frac{u}{3}\right)\right) \sim \exp\left(-\frac{1}{3}pMu\right).$$

Note that this approximation assumes that $u \ll 1$.
We also have:

$$(1 - u)^{pM/3} = \exp(\log((1 - u)^{pM/3})) = \exp(pM/3 \, log(1 - u)) \sim \exp\left(-\frac{1}{3}pMu\right).$$

Hence, we can consider the number of lethal positions $m$ as roughly equal to $pM/3$. Note that this approximation assumes that $u \ll 1$.

## Weighted ERF

If mutants differ in infectivity (mortality or other risks), the ERF can be calculated as a weighted sum over integrated mutant abundances. Assume that $n_1$ mutants have risk $r_1$ and $n_2$ mutants have risk $r_2$.

For virus dynamics, we have

$$\dot{x} = x(bq^{m+n_1+n_2} - a_j)$$

$$\dot{y}_1 = xbq^{m+n_2}(1 - q^{n_1}) + y_1(bq^{m+n_2} - a_j) \tag{38}$$

$$\dot{y}_2 = xbq^{m+n_1}(1 - q^{n_2}) + y_2(bq^{m+n_1} - a_j).$$

Denote by $Y_{i,00}$ the total abundance of mutant $i$ in absence of treatment.
Denote by $Y_{i,01}$ the total abundance of mutant $i$ if treatment starts at peak. The ERF for treatment starting at peak is

$$ERF = \frac{r_1 Y_{1,01} + r_2 Y_{2,01}}{r_1 Y_{1,00} + r_2 Y_{2,00}}. \tag{39}$$

Denote by $Y_{i,11}$ the total abundance of mutant $i$ if treatment starts at infection. The ERF for treatment starting at infection is

$$ERF = \frac{r_1 Y_{1,11} + r_2 Y_{2,11}}{r_1 Y_{1,00} + r_2 Y_{2,00}}. \tag{40}$$

## Supporting information

**S1 Text. Supplementary Figures and Tables.** Supplementary figures: Figs A1 to A31, and supplementary tables: Tables A1 to A4.
(DOCX)

**S2 Text. Step-by-step derivation of Eq 22.**
(PDF)

**S3 Text. Relationship to previous literature and model extensions.**
(DOCX)

## Author Contributions

**Conceptualization:** Gabriela Lobinska, Yitzhak Pilpel, Martin A. Nowak.

**Data curation:** Gabriela Lobinska, Yitzhak Pilpel, Martin A. Nowak.

**Formal analysis:** Gabriela Lobinska, Yitzhak Pilpel, Martin A. Nowak.

**Funding acquisition:** Yitzhak Pilpel.

**Investigation:** Gabriela Lobinska, Yitzhak Pilpel, Martin A. Nowak.

**Methodology:** Yitzhak Pilpel, Martin A. Nowak.

**Project administration:** Yitzhak Pilpel, Martin A. Nowak.

**Supervision:** Yitzhak Pilpel, Martin A. Nowak.

**Validation:** Gabriela Lobinska, Yitzhak Pilpel, Martin A. Nowak.

**Visualization:** Gabriela Lobinska, Yitzhak Pilpel, Martin A. Nowak.

**Writing – original draft:** Gabriela Lobinska, Yitzhak Pilpel, Martin A. Nowak.

**Writing – review & editing:** Gabriela Lobinska, Yitzhak Pilpel, Martin A. Nowak.

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
