## [Editor Report · Decision Letter 0]

16 Sep 2022

Dear Dr Lobinska, 

Thank you for submitting your manuscript entitled "Evolutionary safety of death by mutagenesis" for consideration as a Research Article by PLOS Biology.

Your manuscript has now been evaluated by the PLOS Biology editorial staff, as well as by an academic editor with relevant expertise, and I'm writing to let you know that we would like to send your submission out for external peer review.

Once your full submission is complete, your paper will undergo a series of checks in preparation for peer review. After your manuscript has passed the checks it will be sent out for review. To provide the metadata for your submission, please Login to Editorial Manager (https://www.editorialmanager.com/pbiology) within two working days, i.e. by Sep 20 2022 11:59PM.

Kind regards,

Roli Roberts

Roland Roberts, PhD

Senior Editor

PLOS Biology

rroberts@plos.org

---

## [Decision Letter · Decision Letter 1]

28 Nov 2022

Dear Dr Lobinska,

Thank you for your patience while your manuscript "Evolutionary safety of death by mutagenesis" was peer-reviewed at PLOS Biology. It has now been evaluated by the PLOS Biology editors, an Academic Editor with relevant expertise, and by three independent reviewers. 

You'll see that although the reviewers are intrigued by your findings, they each raise a number of concerns that must be addressed before further consideration. For example, reviewer #1 found your manuscript hard to read and thinks that your study does not take account of the biological mechanisms of lethal mutagenesis, which may limit its informativity (s/he mentions three or four real-life phenomena which would need to be incorporated into the model, if possible). Reviewer #2 starts by questioning an assumption about treatment timing, but then identifies a potential error in the math, and flags further problems with lack of mathematical clarity (it seems that this reviewer may raise further issues when this point is addressed or clarified). Reviewer #3 is positive but raises two points; one (as I understand it) that clinical findings regarding immunosuppressed patients seem to be at odds with your findings, and the other that you should try to use parameters for different SARS-CoV-2 variants (or, if not, run a sensitivity analysis).

In light of the reviews, which you will find at the end of this email, we would like to invite you to revise the work to thoroughly address the reviewers' reports.

Given the extent of revision needed, we cannot make a decision about publication until we have seen the revised manuscript and your response to the reviewers' comments. Your revised manuscript is likely to be sent for further evaluation by all or a subset of the reviewers.

**IMPORTANT - SUBMITTING YOUR REVISION**

*Re-submission Checklist*

*Published Peer Review*

*PLOS Data Policy*

*Blot and Gel Data Policy*

Sincerely,

Roli Roberts

Roland Roberts, PhD

Senior Editor

PLOS Biology

rroberts@plos.org

REVIEWERS' COMMENTS:

Reviewer #1:

In this manuscript Lobinska et al. develop a mathematical model to investigate the balance between efficacy and safety in the use of molnupiravir for SARS-CoV-2. This nucleoside analog is incorporated by the viral polymerase during replication and templates mutations. Given that SARS-CoV-2 and other RNA viruses have mutation rates that are near the maximum tolerable, raising the mutation rate with this drug will reduce viral viability through the accumulation of additional detrimental and lethal mutations. The theoretical downside is that the virus could hit on a mutation that is beneficial - whether through better receptor binding, replication, immune evasion etc. - be transmitted and lead to a new variant of concern. This has been much discussed since the drug's EUA. Sadly, much of the discussion ignores a long history of study of lethal mutagenesis - albeit in other viral systems - which has explored these issues on a theoretical and experimental basis. Some of that work is cited by the authors. The authors develop a model that incorporates various parameters that are known (or at least bounded) for SARS-CoV-2, including, but not limited to: mutation rate, fraction of lethal mutations, growth rate, clearance rate. The parameter space is explored and the boundaries defined where the goals of viral elimination and avoidance of harmful variants are achieved.

As a virologist, and not a mathematical modeler, I found this manuscript a bit dense and difficult to read. This may be how manuscripts in the subfield are written and presented, but it could detract from the readership at a general interest journal at PLOS Biology.

A larger issue is that the model does not account for much of the biology/mechanism of lethal mutagenesis. In this way, it is a bit simplistic in its assumptions and may not really be as informative about the safety and efficacy of lethal mutagenesis as the authors suggest. A few considerations along these lines:

1. Replication mode and number of mutations per generation/cellular infection cycle. The authors should consider the complicating factor of mode of replication. Stamping vs. linear replication (see discussion in cited Sanjuan papers) and whether mutations occur in minus vs. plus strand synthesis can have a profound effect on the number of mutations per genome. See PMID: 25635405. Similarly, RNA editing from APOBEC, which appears to be quite common in SARS-CoV-2, will lead to a higher mutation rate in vivo than the estimates given for the virus passaged in vitro. These issues change the expected number of mutations per genome, perhaps beyond the assumptions of the model.

2. From my reading, the model basically considers the fraction of lethal and non-lethal mutations and the likelihood of the virus making a mutation in either class given its mutation rate. It reads as if any non-lethal mutation is considered potentially beneficial (could lead to a VOC) within the spike RBD or within the rest of the genome either through its direct effects or through establishing a road to additional mutations via higher order epistasis. To me, this ignores some of the recognized complexity. Non-lethal deleterious mutations don't appear to be considered (and also mutations that can reduce fitness through epistasis as well). In the vast majority of situations, these would be outcompeted or cleared faster rather than unmutated wild type viruses. Put another way, the ~60% of mutations with a fitness value of 0.1-0.9 would need to explicitly be considered (see also PMID 27571422 in addition to cited papers from Sanjuan).

3. As in 2, a non-lethal mutation is considered as a candidate for a VOC when it is clear that most VOC emerge after a process that entails a number of mutations arising over a significant period of time (given the known within-host and global rates of evolution). Clearly, the virus would need to hit on the right mix of non-lethal mutations in a short period of time.

4. Other factors that influence the dynamics of mutation accumulation and spread (in VOC and viruses in general) are not considered. First, there is a considerable amount of genetic drift involved in mutations increasing in frequency in vivo (from newly generated mutation to a frequency at which it can plausibly have an effect on phenotype and transmit). Second, through the process of lethal defection (first described in PMID: 15767582), mutated viral genomes can act as dominant negatives and interfere with the replication and progression of non-lethally mutated genomes.

I recognize that some of these factors may be hard to model or incorporate here. However, they are real phenomena and could significantly impact the interpretations and conclusions of the model presented here.

Minor Points

Line 37 - it is unclear why the authors coin a new term "death by mutagenesis" when the term "lethal mutagenesis" has been used by the field for 30 years.

Line 52-53 - while terms like error catastrophe and error threshold are often used in the literature (due to their origins in quasi species theory), these aren't directly applicable to the process of lethal mutagenesis. For models and discussion, see Bull et al. PMID: 17202214

Line 75 - "posology"?

Reviewer #2:

In "Evolutionary safety of death by mutagenesis", authors investigate "evolutionary safety" of drugs whose mechanism of action is to induce mutations during viral replication. 

I was asked to examine specifically the mathematics used in this study. As such I began with the Methods (starting after the references, page 33 of my pdf).

The model (eqs 5a-b) is very simple but in line with models commonly used to get a foothold into such problems. x represents the wild type population, y any and all mutations, and v=x+y the total virus. I am curious about the assumption that the peak time is independence from the treatment - a treatment initiated before peak would affect the peak timing and magnitude, wouldn't dependence on cumulative virus perhaps be more sensible? - but that can wait until the mathematics are corrected.

The first problem is a mistake in the integration in [Disp-formula pbio.3002214.e024]. The answer should be

V- = exp((b*q^m-a1)*T)*exp((b*q^m-a0)*T)/(a1-b*q^m).

The factor exp((b*q^m-a1)*T) is missing.

I don't think that this is a typo b/c the error is repeated, in eqs (12), (13), (14), (16), (17), (18) (note: I stopped after equation 19, as it seemed that the work was built upon incorrect expressions).

The next problem is the material that follows equation (18). It's already incorrect (see above). But then the authors say "clearly, yT=vT-xT". It's not obvious why the definition of a new variable merits a "clearly" but also the significance/utility of this new variable is not clear. The expression of interest at the time is Y-, the cumulative mutants after the infection. It is given as

Y- = vT/(a1-b*q1^m) - xT/(a1-b*q1^(m+n))

I think the authors mean to put the expression on common denominator and yT is meant to be the numerator, but since the denominators of the two terms are different, the numerator is not vT-xT. It's (vT-xT)*a1 - b*q1^m*(vT*q1^n-xt).

So the meaning of yT is not clear (and again, the expression for Y- is not correct, see above).

The authors then investigate the behaviour of Y-(u1) depending on yT (see note above on yT). Further, yT=vT-xT, and the expressions for the latter two isn't correct as a result of the integration error above. But even putting that aside, the conclusions are not obvious. I played with the expressions a bit and did not see where they came from. And if I didn't see it, many other readers won't either. In my opinion the derivation of these expressions needs to be explained, either here in an SI.

Since I'm not confident yT even is the correct expression to be using giving errors detailed above, this is the point at which I stopped. I think the subject matter is interesting and I hope this can be corrected. 

Minor points so far:

(1) Equation (8) is an approximation, the authors neglect 1/(bq^m-a0) which is fine assuming that the viral load at time T is large. But why not explain it, it's an extra line, for clarity? I also note that means that many of the expressions that follow starting with V (eq 12) are also approximations, not exact as is indicated.

(2) Then authors examine behaviour of Y-(u1) depending on yT (notes on that above). Authors should remind readers that u1=1-q1 b/c the equation for Y- does not contain u1 and you have to go back two pages to find it. 

Reviewer #3:

[identifies herself as Pia Abel- zur Wiesch]

The manuscript «Evolutionary safety of death by mutagenesis" is well written and addresses a very important concern regarding the use of the antiviral molnupiravir (lagevrio): that new COVID variants might emerge more easily in treated patients. This is an important and timely contribution- recently, encouraging results of molnupiravir use were published from the Panoramic trial. Moreover, there are only two oral antivirals available. The more widely used paxlovid is difficult to use because of potentially life-threatening interactions in patients receiving multiple other drugs. This limits the use of paxlovid in the most vulnerable groups.

However, I have two major concerns that need to be addressed before I can recommend publication:

- The authors state that molnupiravir use may be safer in individuals with low clearance. This contrasts findings that new mutants arise more easily with long term infections in immunosuppressed patients (e.g. Weigang et al., https://www.nature.com/articles/s41467-021-26602-3) and these patients may have been the origin of the alpha variant (https://www.nature.com/articles/s41586-021-03291-y). It is not immediately apparent to me how exactly low clearance affects the viral replication rate and therefore mutagenesis, and how these different scenarios were fitted to different patient data. Critically ill patients for example can have very high viral loads, e.g. https://www.atsjournals.org/doi/full/10.1164/rccm.202009-3386LE. If the authors assume that the viral load is constant but just the turnover is low, a reduced viral replication rate also leads to a slower accumulation of mutants. If this is true, it must be corrected (using viral load data from immunosuppressed patients) because it fundamentally alters the conclusion. I would expect then that molnupiravir is safer in patients with an intact immune system.

- The authors should clarify which variants their results are valid for. Depending on variant and vaccination status, the time to peak load and clearance can differ https://www.nejm.org/doi/full/10.1056/nejmc2102507. It would be great to obtain sets of parameter estimates for individual variants/vaccine status, and if impossible, this should be stated and a separate sensitivity analysis should be done.

Pia Abel- zur Wiesch

---

## [Decision Letter · Decision Letter 2]

6 Jun 2023

Dear Dr Lobinska,

Thank you for your patience while we considered your revised manuscript "Evolutionary safety of lethal mutagenesis" for publication as a Research Article at PLOS Biology. This revised version of your manuscript has been evaluated by the PLOS Biology editors, the Academic Editor, and one of the original reviewers. However, reviewers #1 and #3 were unable to provide a review in a timely fashion, so we recruited new reviewer #4 to cover the expertise previously addressed by reviewer #1. As you undoubtedly know, I've explained these problems to Dr Pilpel and apologised for the delays incurred, but I'd just like to take this opportunity to apologise to you personally for the extraordinary length of time that this round of review has taken.

Based on the reviews, we are likely to accept this manuscript for publication, provided you satisfactorily address the remaining points raised by reviewer #2. Please also make sure to address the following data and other policy-related requests.

IMPORTANT - please address the following:

a) Your current Title is snappy and intriguing, but a little oblique. Please make it more informative by changing it to: "Evolutionary safety of lethal mutagenesis driven by antiviral drugs"

b) Please address the remaining concerns from reviewer #2.

c) Please provide a blurb, according to the instructions in the submission form.

d) Please address my Data Policy requests below; specifically, we need you to supply the code required to generate Figs 2AB, 3AB, 4ABCD, 5AB, S1ABCD, S2, S3, S4AB, S5ABCDEF, S6ABCDEFGHIJ, S7ABCDEFGHIJ, S8ABCDEFGHIJ, S9ABCDEFGHIJ, S10AB, S11, S12, S13AB, S14AB, S15AB, S16AB, S17AB, S18, S19AB, S20AB, S21AB, S22, S23, S24, S25, S26, S27, S28, S29, S30AB, S31; also (in the additional supplement) S2AB, S3, S4, S5, S6, S7, S8ABCDEFGHIJ, S9ABCDEFGHIJ, either as a supplementary data file or as a permanent DOI’d deposition. We note that you mention a Github deposition (https://github.com/gabriela3001/molnupiravir_evol_safety); this looks very comprehensive, but please clarify whether this can indeed allow readers to reproduce all of the Figures.

e) Because the Github deposition can be changed or deleted at any time, we will need you to make a permanent DOI’d version (e.g. in Zenodo), and to cite this latter URL in the manuscript (see below).

f) Please cite the location of the data/code clearly in all relevant main and supplementary Figure legends, e.g. “The data underlying this Figure can be found in https://doi.org/10.5281/zenodo.XXXXX”

We expect to receive your revised manuscript within two weeks. 

*Published Peer Review History*

*Press*

Sincerely,

Roli Roberts

Roland Roberts, PhD

Senior Editor,

rroberts@plos.org,

PLOS Biology

DATA POLICY:

Regardless of the method selected, please ensure that you provide the code required to generate the following figure panels as they are essential for readers to assess your analysis and to reproduce it: Figs 2AB, 3AB, 4ABCD, 5AB, S1ABCD, S2, S3, S4AB, S5ABCDEF, S6ABCDEFGHIJ, S7ABCDEFGHIJ, S8ABCDEFGHIJ, S9ABCDEFGHIJ, S10AB, S11, S12, S13AB, S14AB, S15AB, S16AB, S17AB, S18, S19AB, S20AB, S21AB, S22, S23, S24, S25, S26, S27, S28, S29, S30AB, S31; also (in the additional supplement) S2AB, S3, S4, S5, S6, S7, S8ABCDEFGHIJ, S9ABCDEFGHIJ. NOTE: the numerical data provided should include all replicates AND the way in which the plotted mean and errors were derived (it should not present only the mean/average values).

DATA NOT SHOWN?

REVIEWERS' COMMENTS:

Reviewer #2:

In "Evolutionary safety of death by mutagenesis", authors investigate "evolutionary safety" of drugs whose mechanism of action is to induce mutations during viral replication. 

I was asked to examine specifically the mathematics used in this study. As such I began with the Methods (starting after the references, page 59 of my pdf).

Thank you to the authors for their response to my previous questions which addressed the concerns I brought up. I've now been through the rest of the methods and have two recommendations and a question.

Recommendations:

(1) Include an SI with the derivations of the mathematical expressions.

I was able to verify most, but not all, of the math. Examples: I don't know how to derive equation (22), and further the parameter regimes aren't clear; when I plug equation (21) into dY-/du I don't recover zero, but it's close for all parameters tested, so I'm guessing it's an approximation. These questions would be clarified with derivations in a separate document. While I'm sure to some top experts such details are unnecessary, if I can't derive them, there are others who won't be able to, either; for reproducibility, in my opinion, these derivations should be included. I don't anticipate this would be too much work for the authors as the calculations are done already. 

(2) When using approximations, please indicate parameter regimes for which the approximations are valid.

In some places, the authors have done as much (e.g. line 904) but not everywhere (e.g. lines 920, 940, others). I'm sure these seem obvious to the authors! But this is an intriguing and potentially important study being published in a biology journal, so it's likely the mathematical expressions will be used by non-modelers blindly. To minimize the risk of bad science downstream, I urge the authors to put the regimes of validity up front.

Finally, my question:

(1) Authors model initiation of immune responses at time T, forcing a peak in the viral load by increasing the clearance rate a so that x'<0 (that is an approximation consistent with some, though not all, hypotheses on viral load peaks). However, when treatment initiated pre-peak, T is kept as an independent parameter. Would treatment - controlling viral loads and therefore immune stimulation by foreign antigen - not delay immune responses' achieving "full strength"? At the very least should T and treatment time not be correlated in some way? Indeed, treatment may be sufficiently effective so that there is no peak (bq1^(m+n)<a0), so T is activation of immune responses only. This is the question I was hinting at in my previous review.

Reviewer #4:

[identifies himself as Raul Andino]

The study conducted by Lobinska, Pilpel and Novak sheds light on the safety and efficacy of antiviral lethal mutagenesis. Their research focuses into the use of nucleoside analogs, a prevalent type of antiviral drugs, that increase the viral mutation rate, causing lethal mutagenesis of the virus. By evaluating the impact of these drugs, the study provides crucial insight into the potential of antiviral lethal mutagenesis as a weapon against viral infections. It presents a comprehensive outlook at the various aspects of this mechanism, including its safety and efficiency, paving the way for future exploration in the field. The results shared in this study will undoubtedly be an essential addition to the existing body of knowledge on antiviral drugs and their application in treating infectious diseases.

As research continues to explore the use of mutagenic treatments for viruses, concerns about their long-term impact on virus evolution have been considered. Namely, their ability to mutate may create an increased number of mutants over time, which could increase virus fitness and pose a safety concern. To address this issue, Lobinska et al have developed a mathematical framework to compare the total mutant load produced with and without mutagenic treatment. This framework considers a variety of variables, such as timing of treatment and patient immune competence, to predict the rates of viable virus mutants. By using realistic assumptions about viral vulnerability and mutation potential, they provide insight into the potential implications of mutagenic treatments on evolutionary safety.

In the wake of the COVID-19 pandemic, many treatments and drugs have been developed to combat the virus. In particular, Molnupiravir has received FDA approval as a viable treatment option. However, it is crucial to consider the potential evolutionary impact of such drugs. Through extensive analysis, researchers have found that Molnupiravir may be narrowly evolutionarily safe, though this is subject to the current estimate of parameters. To further increase evolutionary safety, restricting treatment with this drug to individuals with a low immunological clearance rate may be beneficial. Additionally, future treatments may be designed to lead to a greater increase in the mutation rate to improve the overall evolutionary impact of these drugs. Consideration of these factors is crucial in developing effective treatments while also taking into account potential long-term consequences. In this interesting study, the authors present a new mathematical rule to help determine the fold-increase in mutation rate necessary for pathogen-treatment combinations to achieve evolutionary safety. The model, which is simple but effective, has far-reaching implications for the development of new treatments against a variety of diseases. This report underscores the importance of mathematical models in the field of medical research and highlights the potential for interdisciplinary collaboration between scientists and mathematicians. With this new tool, we can better understand the complex dynamics of pathogen-treatment interactions and develop more effective treatments to combat evolving pathogens.

The level of effort put forth in addressing the concerns of reviewers is a mark of excellence in academic writing, and the author of this article has clearly exemplified that quality. In addition to the original data presented in the model, the author has provided an in-depth analysis of the data to ensure the validity and comprehensiveness of the article. The additional analyses conducted have helped strengthen the original model and presented a more complete picture of the research topic. The resulting article is well supported and should serve as a valuable resource to researchers and scholars in the field. It is reassuring to see such a thorough approach taken, and it is a testament to the author's dedication and expertise in the field.

---

## [Editor Report · Decision Letter 3]

23 Jun 2023

Dear Dr Lobinska,

Thank you for the submission of your revised Research Article "Evolutionary safety of lethal mutagenesis driven by antiviral treatment" for publication in PLOS Biology. On behalf of my colleagues and the Academic Editor, Andrew Read, I'm pleased to say that we can in principle accept your manuscript for publication, provided you address any remaining formatting and reporting issues. These will be detailed in an email you should receive within 2-3 business days from our colleagues in the journal operations team; no action is required from you until then. Please note that we will not be able to formally accept your manuscript and schedule it for publication until you have completed any requested changes.

Sincerely, 

Roli Roberts

Senior Editor

PLOS Biology

rroberts@plos.org